# Short communication: Resolving the discrepancy between U–Pb age estimates for the 'Likhall' bed, a key level in the Ordovician timescale

André N. Paul[1*], Anders Lindskog[2], Urs Schaltegger[1]

1) Department of Earth Sciences, Université de Genève, Switzerland

2) Department of Geology, Lund University, Sweden

*corresponding author, e-mail: Andre.Paul@unige.ch

**Abstract**

The 'Likhall' bed is a rare case of a single-age zircon population from a carbonate rock, which in this case is contextualised with remarkable biotic and environmental changes as well as meteorite bombardment of Earth after an asteroid breakup in

space. Published chemical-abrasion, high-precision isotope-dilution, thermal ionization mass spectrometry (CA-ID-TIMS) U–Pb age estimates disagree at the typical precision of <0.1% for a $^{206}Pb/^{238}U$ date, which has led to discrepancies in the interpretation of the timing of events and their possible cause–effect relationships. We evaluate here the relative strengths and weaknesses, and discrepancies in the so far published datasets, propose strategies to overcome them and present a new U-Pb dataset with improved precision and accuracy. Ultimately, we find that domains of residual Pb loss are a significant source of

age-offset between previously published data, amplified by differences in data evaluation strategies. Our new dataset benefits from an improved chemical abrasion protocol resulting in a more complete mitigation of decay-damage induced grain portions, and points to a weighted mean age estimate of 466.37±0.14/0.18/0.53 Ma for the 'Likhall' zircon population. This age is intermediate between previous estimates, but outside of analytical uncertainty, and provides a firm tie point for the Ordovician timescale.

## 1 Introduction

The Ordovician 'Likhall' bed hosts zircons, fossils and is an 'orthoceratite limestone' outcropping at Kinnekulle, Sweden, locally referred to as 'Täljsten', which is an important marker interval in the regional stratigraphy (see below). These strata record remarkable changes in paleoenvironmental conditions and biodiversity (Lindskog and Eriksson, 2017; Servais and Harper, 2018; Lindskog et al., 2017). Besides being unusually rich in prismatic zircon of apparently single age, the 'Likhall'

bed coincides with an interval with uniquely abundant L-chondritic 'fossil' meteorites and purportedly related chromite grains (Lindskog et al., 2017; Liao et al., 2020; and references therein). This has been linked to the breakup of an asteroid in space and temporal overlap between excess meteoritic matter and prominent climatic changes and biodiversity peaks has spurred the controversial hypotheses that meteorite bombardment fundamentally affected the Ordovician climate and instigated biodiversification (Schmitz et al. 2008, 2019; cf. Lindskog et al. 2017; Rasmussen et al. 2021). Considering absolute/numerical

time perspectives two individual U–Pb geochronological studies have arrived at significantly different age estimate of zircon from the 'Likhall' bed – 467.50±0.28 (Lindskog et al. 2017) and 465.18±0.17 Ma (Liao et al. 2020) – each reaching contrasting conclusions for the absolute timing of the L-chondrite breakup event and with differing implications for the Ordovician event stratigraphy and timescales (particularly the Darriwilian Stage).

While the occurrence of an apparently single U–Pb zircon age population in a carbonate rock is already uncommon, the percent difference between these two previously published $^{206}Pb$-$^{238}U$ ages amounts to 0.4%, which exceeds the expected reproducibility level of, for example, natural zircon reference materials (0.1%; Schaltegger et al., 2021). Thus, we have to evaluate the accuracy of the published U–Pb ages on the 'Likhall' zircons before we can determine a more robust age estimate. A significant effort has recently been undertaken by the U-Pb community to reduce inter-lab bias in isotope dilution – thermal ionisation mass spectrometry (ID-TIMS), via the use of precisely calibrated EARTHTIME (ET) isotopic tracers (Condon et al., 2015; McLean et al., 2015), community-wide shared data reduction (Bowring et al., 2012; McLean et al., 2011) and sample preparation procedures (Widmann et al., 2019). However, the two previously published 'Likhall' zircon datasets were produced employing different isotopic tracers, instrumentation and chemical abrasion procedures: A mixed ET $^{202}Pb$-$^{205}Pb$-$^{233}U$-$^{235}U$ tracer was used in the Lindskog et al. (2017) study, while it was an in-house $^{202}Pb$-$^{205}Pb$-$^{233}U$-$^{236}U$ tracer for the Liao et al. (2020) study. Further differences concern the measurement procedure of the U isotope composition (on TIMS as $UO_2$ for the first, on a multi collector – inductively coupled plasma – mass spectrometer (MC-ICP-MS) as a metal for the latter). Most importantly, chemical abrasion procedures also differ, being annealing at 900°C for 3 days and leaching at 180°C for 12hrs (Lindskog et al., 2017) and annealing at 900°C for 48h and leaching at 190°C for 15hrs (Liao et al., 2020).

The variation of the chemical abrasion procedure parameters has profound effects on the U–Pb zircon isotopic systematics. The duration as well as the temperature of the partial dissolution may have significant impact on the potential to effectively remove structurally damaged domains, which typically produce anomalously young dates (Mattinson, 2005; Huyskens et al., 2016; Keller et al., 2019; Widmann et al., 2019). Both of the aforementioned studies of the 'Likhall' bed utilised temperature lower than the most recent recommendation for the chemical abrasion procedure (McKanna et al., 2023a, 2023b; Widmann et al., 2019), which raises concerns about remnant Pb loss domains present in the zircons analysed. Different chemical abrasion procedures may be variably effective in removing visible inclusions (e.g. apatite and/or melt inclusions) in the analysed zircon crystals.

The impact of tracer uncertainty is significantly lower than the 0.4% discrepancy between the absolute ages obtained from 'Likhall' bed. Similarly, the variation of the instrumental setup for U isotope analysis does not seem to introduce significant off-sets, although we cannot evaluate this with certainty, due to limited amount of comparable data. The $^{206}Pb/^{238}U$ age of the natural reference zircon material Temora reported by Liao et al. (2020) from the mixed TIMS–MC-ICP-MS analysis (417.19±0.15 Ma) is in agreement with the most recent estimates of 417.310 ± 0.074 Ma (von Quadt et al., 2016) and 417.353±0.052 Ma (Schaltegger et al., 2021). Reported values of the ET100 solution from the lab utilised in the Liao et al. (2020) is 100.169±0.047 Ma, indistinguishable from the reference value of 100.173±0.003 Ma (Schaltegger et al., 2021) and our value produced (100.1678 ± 0.0046 Ma) during the period of data collection of this study.

Thus, we can reasonably assume that the difference in the chemical abrasion procedure is the main source for the significant difference between the two previously published studies of 'Likhall' zircons. Furthermore, there is a difference in data interpretation strategy of the two published datasets. Lindskog et al. (2017) put the emphasis on the largest, statistically valid, weighted mean age plateau, whereas Liao et al. (2020) chose the youngest cluster of zircon U–Pb analyses. In the following, we will first explore different interpretation strategies of the existing data, such as *i)* looking for the largest statistically valid plateau, *ii)* the youngest cluster, *iii)* the duration of time recorded by the youngest and oldest zircon U–Pb date ($\Delta$t) and *iv)* the youngest concordant single zircon U–Pb analysis. Ideally, applying the same interpretation methodology to both datasets should reduce the discrepancy of the absolute U–Pb ages. Furthermore, we add another set of U-Pb dates from the same material but apply the chemical abrasion procedure of Widmann et al. (2019), which should more effectively mitigate Pb loss. Ultimately, we will suggest a revised age for the 'Likhall' bed.

## 2 Methods

Single grains of 'Likhall' zircon crystals free of visible inclusions and cracks were hand-picked using a binocular microscope at a magnification of ×20 to ×40 from the same petri dish as the material analyzed by Lindskog et al. (2017). The size of individual zircons was variable, with length ranging from ~50µm to ~300µm. The chemical abrasion procedures were 48hrs of annealing at 900°C, 12hrs partial dissolution at 210°C, as defined as optimal in Widmann et al. (2019). Afterwards, we observe that some zircons had either fragmented or embayment's at the core predominantly, or some channels had more efficiently progressed inwards, without any systematic preference identifiable. Individual zircon grains were washed in 3ml Savillex beakers in an ultrasonic bath, 4× in 7N HNO$_3$, transferred into individual 200µL Savillex microcapsules, along with 2–3 drops of HF$_{conc}$ and 3.9–5.5 mg of a mixed $^{202}$Pb-$^{205}$Pb-$^{233}$U–$^{235}$U tracer solution (ET2535, Condon et al., 2015; McLean et al., 2015), and dissolved at 210°C in a Parr bomb for 48hrs. After dissolution, samples were dried down on a hotplate at 120°C, re-dissolved in 3N HCl, and then U and Pb were separated using a single column anion exchange chemistry. Uranium and Pb were loaded on outgassed, zone-refined, single Re filaments with a silica-gel/phosphoric acid emitter solution (Gerstenberger and Haase, 1997).

Uranium and Pb isotopic compositions were measured on an IsotopX Phoenix TIMS at the University of Geneva. Lead was measured in dynamic mode using a Daly detector, U was measured as an oxide in static mode using Faraday cups coupled to $10^{12}\,\Omega$ resistance amplifiers. Measured isotopic ratios were corrected for interferences of $^{238}$U$^{18}$O$^{16}$O on $^{235}$U$^{16}$O$_2$ using a $^{18}$O/$^{16}$O composition of 0.00205, based on repeat measurements of the U500 standard. Mass fractionation of U was corrected using a double isotope tracer with a $^{235}$U/$^{233}$U of 0.99506±0.005. Mass fractionation of Pb is calculated and corrected using a $^{202}$Pb/$^{205}$Pb ratio of 0.99924±0.00027 (1$\sigma$) (Condon et al., 2015). Zircon Pb analyses were corrected for laboratory blanks, with a $^{206}$Pb/$^{204}$Pb of 17.10±0.21, a $^{207}$Pb/$^{204}$Pb of 15.07±0.11 and a $^{208}$Pb/$^{204}$Pb of 36.17±0.25 (all 1$\sigma$), based on repeat measurements of total procedural blanks for the zircon U-Pb column chemistry. All data were processed using the Tripoli v. 4.10 and Redux v. 3.7.1 U-Pb software (Bowring et al., 2012; McLean et al., 2011). Weighted mean U-Pb age uncertainties are reported at the

2σ level in the format A±X/Y/Z, where A is the weighted mean age, X is analytical uncertainty, Y is analytical and tracer uncertainty combined, and Z is analytical, tracer, and decay constant uncertainties combined (Schoene et al., 2006). Thorium abundances for each grain, and subsequently Th/U$_{zircon}$, were calculated using the abundance of $^{208}$Pb within the crystal and the $^{206}$Pb/$^{238}$U age to calculate radiogenic in-growth (McLean et al., 2011). All U-Pb dates are corrected for initial $^{230}$Th disequilibrium, assuming a Th/U ratio of the magma of 3.5±1.

Repeat analyses of the ET100 solution ($^{206}$Pb/$^{238}$U date: 100.173 ± 0.003 Ma; Schaltegger et al., 2021) yielded a value of 100.1678 ± 0.0046 Ma (MSWD = 3.2, n = 32/40), during the period of data collection. The eight rejected ET100 samples were prepared and analysed consecutively, , yielding an anomalously young average age, a detail observed by previous studies (Schaltegger et al., 2021).

## 3 Geological Setting

The Baltoscandian region was largely covered by an epeiric sea throughout much of the Ordovician, leaving behind a sedimentary record mainly in the form of mudstones, shales, and limestones (e.g., Nielsen et al., 2023, and references therein). In the Lower–Middle Ordovician, the continent Baltica was at a relatively high latitudinal position and any relief of the regional Precambrian basement was near-completely smoothed out. These conditions limited terrigenous input into the basin, and sedimentation rates were typically low (a few mm/ka). The regional intra-cratonic Ordovician succession is therefore relatively thin. Volcanic dust may have contributed a relatively large proportion of the non-carbonate sedimentary materials (Lindstrom, 1974). The Middle Ordovician (Dapingian–Darriwilian) was mainly characterized by widespread deposition of cool-water carbonate sediments, in Sweden commonly referred to as the 'orthoceratite limestone' (e.g., Lindskog and Eriksson, 2017; Lindskog et al., 2023).

The Ordovician succession of Baltoscandia contains numerous bentonite beds of varying thickness and lateral distribution, the most prominent of which occur in the Upper Ordovician (e.g., Ballo et al., 2019; Bergström, 1989; and references therein). Few of the bentonite beds have been isotopically dated, and even fewer so using modern techniques such as CA-ID-TIMS. Some carbonate beds in the 'orthoceratite limestone' contain abundant zircon, and the crystal characteristics and U–Pb age data of the grains indicate a volcanic origin (Lindskog et al. 2017; Liao et al. 2020). Thus, the zircon-rich beds arguably represent 'crypto-tephra' (McLaughlin et al., 2023).

The table-mountain Kinnekulle in the province of Västergötland, south-central Sweden, hosts a relatively expanded 'orthoceratite limestone' succession (e.g., Lindskog and Eriksson, 2017). These rocks have been the target of several studies of paleontology, sedimentology, and geochemistry (e.g., Ahlberg et al., 2023, and references therein). Our sample materials derive from the Thorsberg quarry on eastern Kinnekulle (WGS84 coordinates 58.579167, 13.429444), from a distinct, gray-colored interval traditionally referred to as the 'Täljsten' ('carving stone') by local quarrymen. In more detail, the sample derives from a specific bed referred to as 'Likhall' ('corpse slab'), which has been shown to contain very abundant zircon

grains (Lindskog et al. 2017; Liao et al. 2020). The biostratigraphic context of this c. 10 cm thick bed is very well known, and its base coincides with that of the geographically widely distributed *Yangtzeplacognathus crassus* conodont Zone in the middle Darriwilian (for more details, see Lindskog et al. 2017).

## 4 Results

A total of 22 zircons from the zircon-rich 'Likhall' bed were analysed for their U–Pb isotopic composition. The U and Pb isotopic data are presented in Table 1, interpreted U–Pb dates are illustrated in concordia space in Fig. 1, revealing a large spread in $^{206}Pb/^{238}U$ dates as well as variable discordance. The new data presented here comprise 15 analytically concordant and 7 discordant $^{206}Pb/^{238}U$ zircon analyses, which range in $^{206}Pb/^{238}U$ dates from 468.8±1.2 to 462.43±0.27 Ma. The ratio of radiogenic to common lead (Pb*/Pb$_c$) ranges from 11.1 to 79.7, with variable amounts of Pb$_c$ (0.20 to 1.09 pg). The $^{207}Pb/^{206}Pb$ dates range from 461±10 to 490±15 Ma.

The compilation of previously published zircon U-Pb analyses of the 'Likhall' bed comprises 17 analyses from Lindskog et al. (2021) and 21 analyses from Liao et al. (2020), which are now complemented by our 22 new analyses in this study. The $^{206}Pb/^{238}U$ dates of all these datasets are plotted in Fig. 2, along with the weighted mean $^{206}Pb/^{238}U$ age for each of the datasets following the selection criteria discussed below. All data are available in Table 1 and Supplement (Table S1).

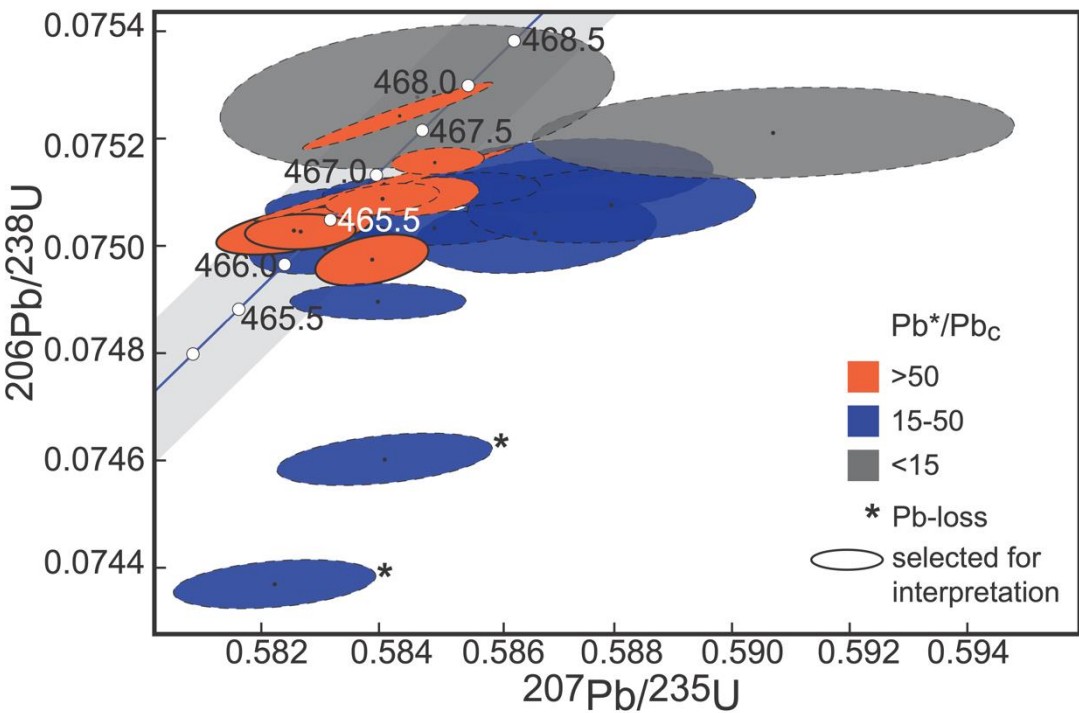

**Fig. 1) Concordia diagram generated in ETRedux v. 3.7.1, using colour coding for bands of Pb\*/Pb$_c$ as discussed in the text for the error ellipses. Orange colour indicates higher Pb\*/Pb$_c$ >50, whereas blue colour indicates Pb\*/Pb$_c$ values between 15-50, grey error ellipses indicate Pb\*/Pb$_c$ <15. Ellipses marked by an asterisk are interpreted to be affected by Pb loss. Black outline indicates analyses considered for the weighted mean age, where the selection criterion is Pb\*/Pbc >50 and analytical concordance (part or full ellipse overlaps with the uncertainty band of the concordia curve). Grey band illustrates the uncertainty band of the concordia curve.**

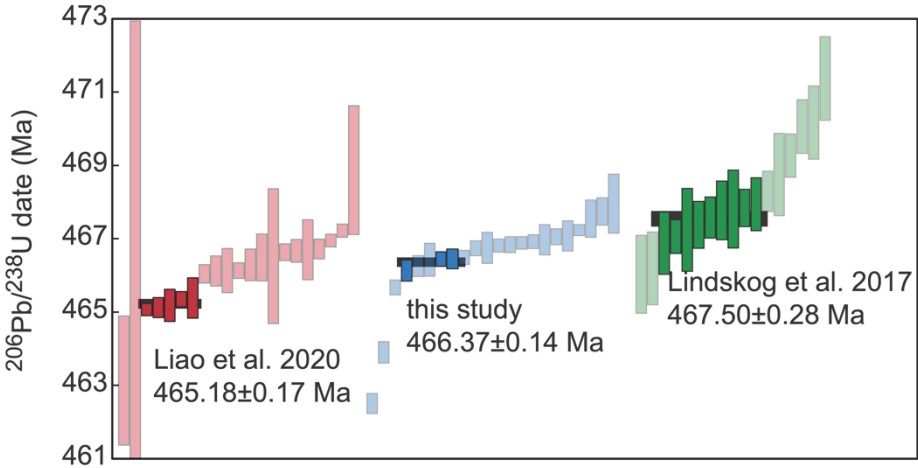


**Fig. 2) Rank order plot of calculated [206]Pb/[238]U/ single zircon U-Pb dates from Liao et al (2020), this study and Lindskog et al (2017). Transparent bars were not considered for the calculation of the weighted mean age, as per each studies selection criteria (see text for more details).**

 **Tab. 1) U-Pb ID-TIMS measurement results of individual zircons from the ´Likhall´ bed.**

Tab. 1) U-Pb ID-TIMS measurement results of individual zircons from the 'Likhall' bed.

| Zircon | No. of Grains | Dates (Ma) 206Pb/238U <Th> a | ±2σ abs | 206Pb/238U b | ±2σ abs | 207Pb/235U <Pa> c | ±2σ abs | 207Pb/235U b | ±2σ abs | 207Pb/206Pb <Th> a | ±2σ abs | 207Pb/206Pb b | ±2σ abs | % disc d | Composition Pb* (pg) e | Pbc (pg) f | Pb*/Pbc g | Th/U h | Isotopic Ratios 206Pb/204Pb i | 208Pb/206Pb j | 206Pb/238U i | ±2σ % j | 207Pb/235U | ±2σ % | 207Pb/206Pb | ±2σ % | Corr. coef. |
|---|---|---|---|---|---|---|---|---|---|---|---|---|---|---|---|---|---|---|---|---|---|---|---|---|---|---|---|
| **Likhall** | | | | | | | | | | | | | | | | | | | | | | | | | | | |
| Likhall_z14 | 1 | 462.51 | 0.27 | 462.43 | 0.27 | 465.9 | 1.1 | 465.9 | 1.1 | 482.6 | 6.2 | 483.0 | 6.2 | 4.3 | 19.7 | 0.59 | 33.2 | 0.78 | 1798 | 0.24 | 0.0744 | 0.06 | 0.5822 | 0.30 | 0.0568 | 0.28 | 0.33 |
| Likhall_z18 | 1 | 463.90 | 0.29 | 463.82 | 0.29 | 467.1 | 1.2 | 467.1 | 1.2 | 482.8 | 6.5 | 483.2 | 6.5 | 4.0 | 24.3 | 0.75 | 32.5 | 0.83 | 1738 | 0.26 | 0.0746 | 0.07 | 0.5841 | 0.31 | 0.0568 | 0.29 | 0.40 |
| Likhall_z17 | 1 | 465.67 | 0.20 | 465.59 | 0.20 | 467.0 | 1.0 | 467.0 | 1.0 | 473.7 | 5.7 | 474.1 | 5.7 | 1.8 | 24.9 | 0.63 | 39.7 | 0.74 | 2165 | 0.23 | 0.0749 | 0.04 | 0.5840 | 0.26 | 0.0566 | 0.26 | 0.01 |
| Likhall_z6 | 1 | 466.13 | 0.29 | 466.05 | 0.29 | 466.9 | 0.6 | 467.0 | 0.6 | 471.0 | 3.4 | 471.4 | 3.4 | 1.1 | 42.7 | 0.85 | 50.0 | 0.81 | 2678 | 0.25 | 0.0750 | 0.06 | 0.5839 | 0.16 | 0.0565 | 0.15 | 0.37 |
| Likhall_z1 | 1 | 466.25 | 0.28 | 466.18 | 0.28 | 466.4 | 0.8 | 466.4 | 0.8 | 467.4 | 4.7 | 467.7 | 4.7 | 0.3 | 43.3 | 1.08 | 40.0 | 0.94 | 2080 | 0.29 | 0.0750 | 0.06 | 0.5831 | 0.22 | 0.0564 | 0.21 | 0.29 |
| Likhall_z8 | 1 | 466.43 | 0.45 | 466.35 | 0.45 | 468.7 | 1.3 | 468.7 | 1.3 | 480.0 | 7.5 | 480.4 | 7.5 | 2.9 | 20.7 | 0.95 | 21.8 | 0.83 | 1172 | 0.26 | 0.0750 | 0.10 | 0.5867 | 0.35 | 0.0567 | 0.34 | 0.25 |
| Likhall_z16 | 1 | 466.44 | 0.20 | 466.37 | 0.20 | 466.2 | 0.6 | 466.2 | 0.6 | 464.8 | 3.6 | 465.2 | 3.6 | -0.2 | 20.6 | 0.30 | 69.1 | 0.90 | 3617 | 0.28 | 0.0750 | 0.06 | 0.5827 | 0.16 | 0.0564 | 0.16 | 0.10 |
| Likhall_z26 | 1 | 466.46 | 0.27 | 466.38 | 0.27 | 466.1 | 0.8 | 466.1 | 0.8 | 464.4 | 4.5 | 464.8 | 4.5 | -0.3 | 19.1 | 0.32 | 58.8 | 0.81 | 3149 | 0.25 | 0.0750 | 0.06 | 0.5826 | 0.23 | 0.0563 | 0.20 | 0.50 |
| Likhall_z11 | 1 | 466.49 | 0.19 | 466.41 | 0.19 | 467.6 | 1.0 | 467.6 | 1.0 | 473.2 | 5.6 | 473.6 | 5.6 | 1.5 | 22.6 | 0.76 | 29.8 | 0.77 | 1614 | 0.24 | 0.0750 | 0.04 | 0.5849 | 0.25 | 0.0566 | 0.25 | 0.08 |
| Likhall_z7 | 1 | 466.74 | 0.20 | 466.65 | 0.20 | 466.5 | 0.7 | 466.5 | 0.7 | 465.2 | 4.0 | 465.6 | 4.0 | -0.2 | 37.6 | 0.93 | 40.6 | 0.60 | 2293 | 0.19 | 0.0751 | 0.04 | 0.5831 | 0.19 | 0.0564 | 0.18 | 0.21 |
| Likhall_z4 | 1 | 466.75 | 0.42 | 466.66 | 0.42 | 469.6 | 1.6 | 469.6 | 1.6 | 483.3 | 9.0 | 483.7 | 9.0 | 3.5 | 15.1 | 0.90 | 16.7 | 0.50 | 974 | 0.16 | 0.0751 | 0.09 | 0.5880 | 0.42 | 0.0568 | 0.41 | 0.20 |
| Likhall_z28 | 1 | 466.81 | 0.18 | 466.73 | 0.18 | 467.1 | 0.6 | 467.1 | 0.6 | 468.3 | 3.4 | 468.7 | 3.4 | 0.4 | 16.2 | 0.20 | 79.7 | 0.79 | 4278 | 0.25 | 0.0751 | 0.04 | 0.5841 | 0.17 | 0.0564 | 0.15 | 0.41 |
| Likhall_z22 | 1 | 466.84 | 0.22 | 466.76 | 0.22 | 467.5 | 0.6 | 467.5 | 0.6 | 470.9 | 3.3 | 471.3 | 3.3 | 1.0 | 25.1 | 0.45 | 55.7 | 0.83 | 2960 | 0.26 | 0.0751 | 0.05 | 0.5848 | 0.16 | 0.0565 | 0.15 | 0.32 |
| Likhall_z13 | 1 | 466.88 | 0.17 | 466.80 | 0.17 | 467.1 | 0.7 | 467.1 | 0.7 | 468.0 | 3.8 | 468.4 | 3.8 | 0.3 | 39.3 | 0.86 | 45.8 | 0.74 | 2496 | 0.23 | 0.0751 | 0.04 | 0.5841 | 0.17 | 0.0564 | 0.17 | 0.21 |
| Likhall_z10 | 1 | 466.92 | 0.19 | 466.84 | 0.19 | 468.0 | 0.8 | 468.0 | 0.8 | 473.4 | 4.4 | 473.8 | 4.4 | 1.5 | 32.6 | 0.87 | 37.6 | 0.76 | 2042 | 0.24 | 0.0751 | 0.04 | 0.5855 | 0.20 | 0.0566 | 0.20 | 0.19 |
| Likhall_z27 | 1 | 466.97 | 0.41 | 466.90 | 0.41 | 467.1 | 1.4 | 467.1 | 1.4 | 467.6 | 6.4 | 468.0 | 6.4 | 0.2 | 14.6 | 0.20 | 71.5 | 0.88 | 3760 | 0.27 | 0.0751 | 0.09 | 0.5841 | 0.38 | 0.0564 | 0.29 | 0.99 |
| Likhall_z24 | 1 | 467.05 | 0.22 | 466.97 | 0.22 | 468.6 | 0.9 | 468.6 | 0.9 | 476.2 | 5.1 | 476.6 | 5.1 | 2.0 | 13.6 | 0.32 | 42.2 | 0.69 | 2329 | 0.22 | 0.0751 | 0.05 | 0.5864 | 0.25 | 0.0566 | 0.23 | 0.55 |
| Likhall_z3 | 1 | 467.07 | 0.41 | 466.99 | 0.41 | 469.1 | 1.6 | 469.1 | 1.6 | 478.9 | 9.2 | 479.3 | 9.2 | 2.6 | 16.2 | 0.93 | 17.4 | 0.73 | 958 | 0.23 | 0.0751 | 0.09 | 0.5872 | 0.42 | 0.0567 | 0.41 | 0.18 |
| Likhall_z12 | 1 | 467.23 | 0.15 | 467.16 | 0.15 | 467.7 | 0.5 | 467.7 | 0.5 | 469.8 | 2.9 | 470.2 | 2.9 | 0.6 | 59.8 | 0.98 | 61.0 | 0.85 | 3227 | 0.27 | 0.0752 | 0.03 | 0.5850 | 0.13 | 0.0565 | 0.13 | 0.21 |
| Likhall_z9 | 1 | 467.55 | 0.51 | 467.47 | 0.51 | 471.3 | 2.6 | 471.3 | 2.6 | 489.7 | 15.1 | 490.1 | 15.1 | 4.6 | 12.1 | 1.09 | 11.1 | 0.88 | 597 | 0.27 | 0.0752 | 0.11 | 0.5907 | 0.69 | 0.0570 | 0.68 | 0.16 |
| Likhall_z25 | 1 | 467.74 | 0.37 | 467.66 | 0.37 | 467.2 | 1.0 | 467.2 | 1.0 | 464.7 | 4.4 | 465.1 | 4.4 | -0.5 | 14.0 | 0.26 | 53.3 | 0.75 | 2893 | 0.23 | 0.0752 | 0.08 | 0.5843 | 0.28 | 0.0563 | 0.19 | 0.97 |
| Likhall_z2 | 1 | 467.95 | 0.80 | 467.87 | 0.80 | 467.4 | 2.1 | 467.4 | 2.1 | 464.9 | 12.1 | 465.3 | 12.1 | -0.5 | 14.2 | 1.02 | 14.0 | 0.80 | 763 | 0.25 | 0.0753 | 0.18 | 0.5846 | 0.57 | 0.0564 | 0.55 | 0.29 |

a  Corrected for initial Th/U disequilibrium using radiogenic $^{208}$Pb and Th/U[magma] = 3.5±1.

b  Isotopic dates calculated using $\lambda^{238}$ = 1.55125E$^{-10}$ (Jaffey et al. 1971) and $\lambda^{235}$ = 9.8485E$^{-10}$ (Jaffey et al. 1971).

c  Corrected for initial Pa/U disequilibrium using initial fraction activity ratio [$^{231}$Pa]/[$^{235}$U] = 1.1.

d  % discordance = 100 – (100 * ($^{206}$Pb/$^{238}$U date) / ($^{207}$Pb/$^{206}$Pb date))

e  Total mass of radiogenic Pb.

f  Total mass of common Pb.

g  Ratio of radiogenic Pb (including $^{208}$Pb) to common Pb.

h  Th contents calculated from radiogenic $^{208}$Pb and $^{230}$Th-corrected $^{206}$Pb/$^{238}$U date of the sample, assuming concordance between U-Pb Th-Pb systems.

i  Measured ratio corrected for fractionation and spike contribution only.

j  Measured ratios corrected for fractionation, tracer and blank.

## 5 Discussion

### 5.1 Radiogenic Pb/common Pb ratio (Pb*/Pb$_c$) as a selection criterion:

The color-coding of the data ellipses in Fig. 1 indicates that the majority of the analyses plot onto or near concordia within its uncertainty band. Analyses with low Pb*/Pb$_c$ (<50) have significantly higher scatter in both $^{206}$Pb/$^{238}$U and $^{207}$Pb/$^{235}$U ratios. Subsequently, we only use high Pb*/Pb$_c$ (>50) analyses for our age interpretation and will also apply this strategy to the previously published datasets. The accuracy of $^{206}$Pb/$^{238}$U dates is higher at Pb*/Pb$_c$ >15 (Schaltegger et al., 2021), which is the case for most of our analyses that underwent chemical abrasion for 12hrs at 210°C, and for some of the analyses of Liao

et al. (2020) abraded for 15hrs at 190°C (Fig. 3). Contrastingly, a major part of the Lindskog et al. (2017) dates abraded for 12hrs at 180°C plot below this threshold at variable $^{206}$Pb/$^{238}$U dates. Considering the significant differences in the range of Pb*/Pbc reported in the three datasets, we have also evaluated Pb*/Pbc bands ranging <15, 15-50 in addition to >50 discussed above.

When considering only analyses with Pb*/Pbc <15, the Lindskog et al (2017), Liao et al. (2020) and this study datasets would

result in weighted mean ages of 468.40±0.50 Ma (MSWD = 2.7, n = 10), 465.92±0.21 Ma (MSWD = 3.6, n = 14) and 467.67±0.84 Ma (MSWD = 0.18, n = 2). Improvement of the weighted mean ages between the studies can be observed when considering Pb*/Pbc band 15-50, where the studies result in 467.12±0.59 Ma (MSWD = 0.46, n = 6), 466.68±0.18 Ma (MSWD = 5.4, n = 6) and 466.79±0.15 Ma (MSWD = 1.4, n = 11) (discordant data not considered, Table S1). In our new dataset, the discrepancy between the proposed U-Pb age when comparing Pb*/Pbc bands 15-50 and >50 is 0.09%, which is the expected

reproducibility of natural reference materials (Schaltegger et al. 2021). Potential sources of this discrepancy may be i) residual Pb loss, ii) inaccurate Pb-blank correction iii) natural variability and duration of magmatic zircon growth in a magma chamber. Where possible, high Pb*/Pbc data should be preferred for interpretation, to reduce effects from potential i) residual Pb loss and ii) inaccurate Pb-blank correction.

Adopting the Pb*/Pb$_c$ (>50 in this study) as a criterion for data selection thus is an important step towards greater accuracy.

The data of Lindskog et al. (2017) shows significant correlation between Pb* and Pb$_c$ (Fig. 4), at overall high Pb$_c$ values (up to ~6 pg), this may indicate that some indiscernible mineral inclusions were not removed during chemical abrasion. Chemical abrasion at higher temperature should access and dissolve these (McKanna et al., 2023a), if the hypothesis of un-resolved inclusions is correct, and would reduce the total Pb$_c$ observed. Subsequently, we would then use only high Pb*/Pb$_c$ (>50) analyses to make an age interpretation.

The high Pb$_c$ concentrations of the Lindskog et al. (2017) analyses (up to 6 pg), suggest that the conditions during the chemical abrasion procedure were not sufficient to remove the inclusions that are clearly visible in the grain separate (Lindskog et al., 2017), leading to generally low Pb*/Pb$_c$. The higher dependency on the Pb$_c$ correction can potentially result in over- or under-correction, which results in younger or older dates calculated. This is further explored in section 5.3 further on.

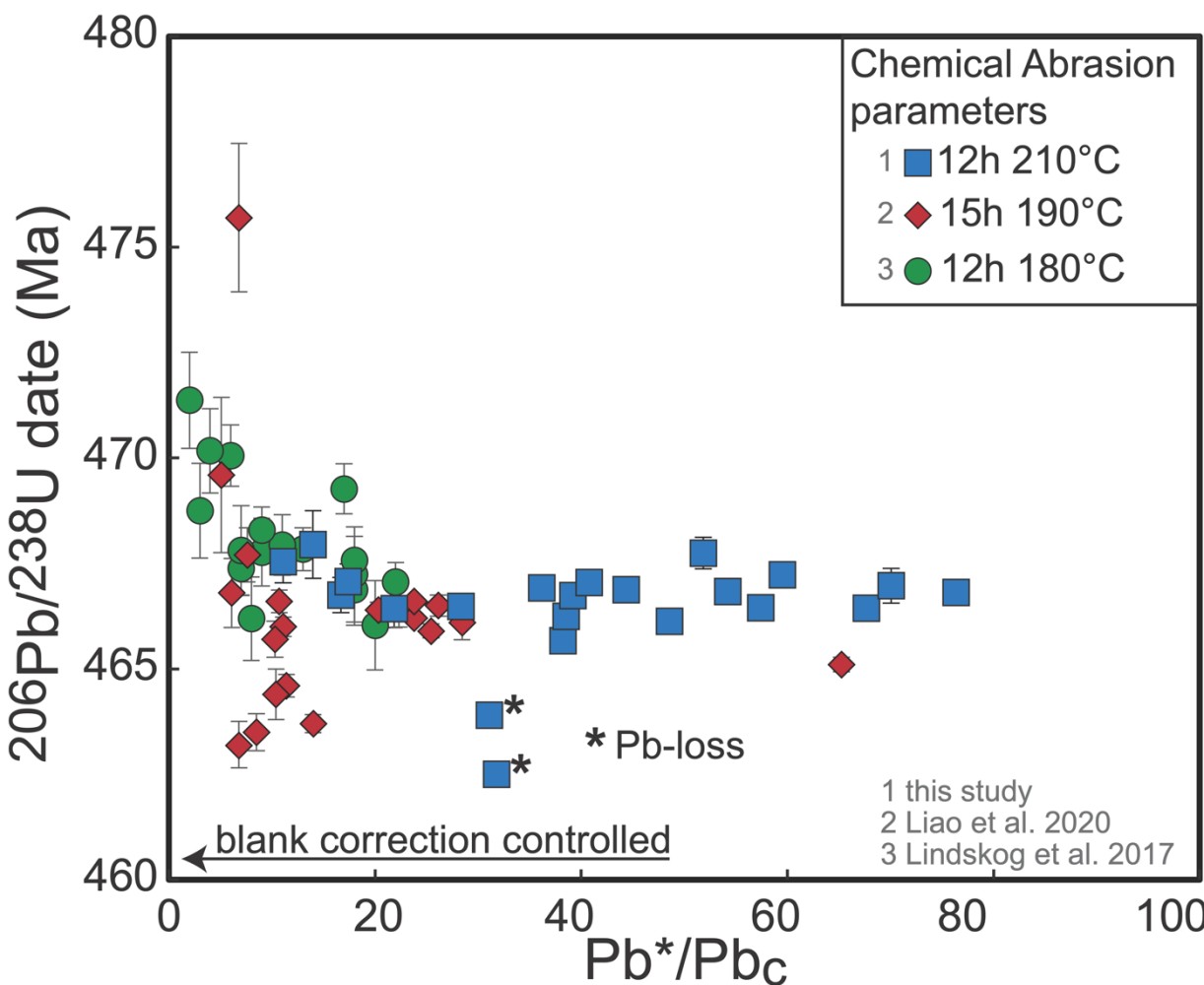

**Fig. 3) Comparison of calculated $^{206}$Pb/$^{238}$U zircon dates with their respective Pb\*/Pb$_c$ values. Low Pb\*/Pb$_c$ values typically indicate stronger effects of blank correction on the calculated $^{238}$U/$^{206}$Pb dates (Schaltegger et al. 2021). Colours indicate the different studies and differences in their chemical abrasion procedure. Samples marked with an asterisk are discordant in concordia space (see Fig. 1).**

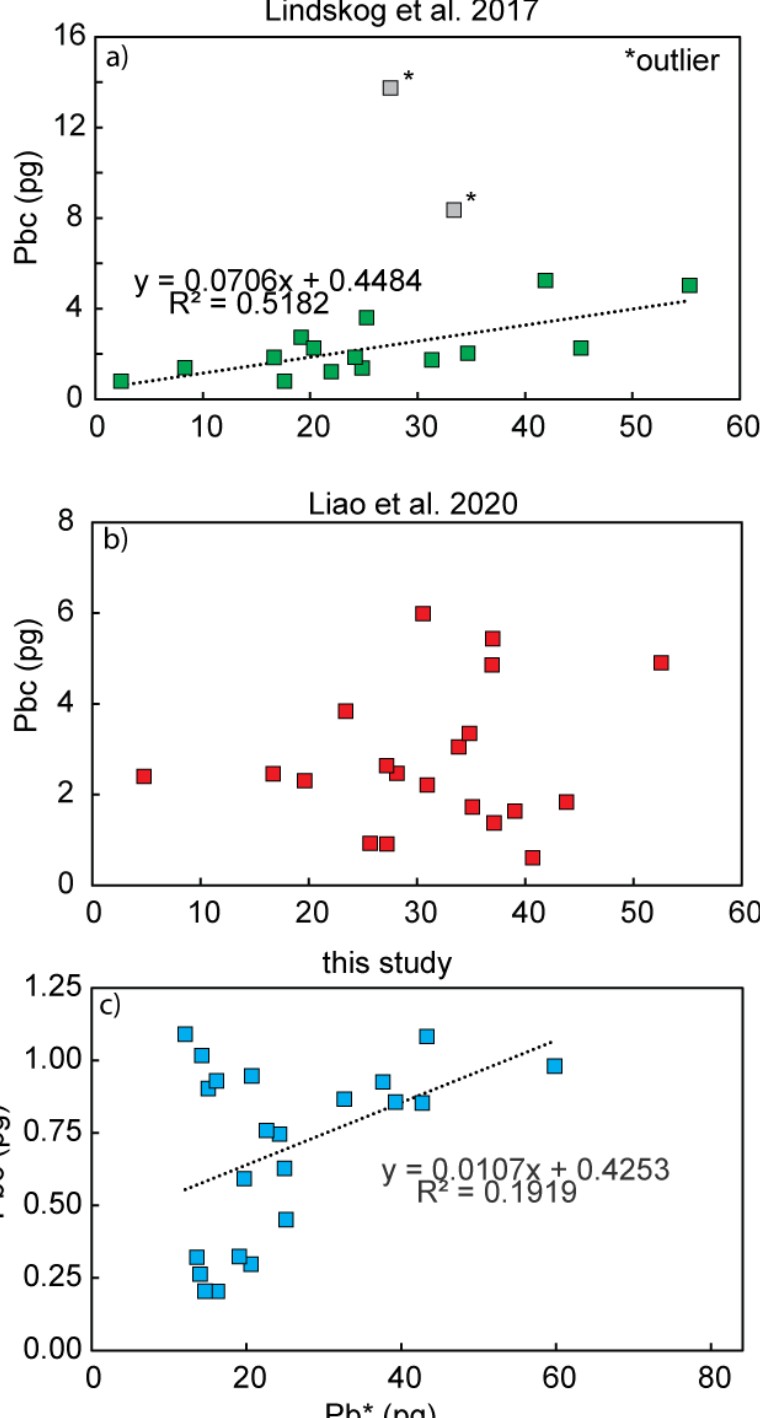


**Fig. 4)** Comparison of $Pb_c$ vs $Pb^*$ of a) the Lindskog et al. (2017) study, b) the Liao et al. (2020) study and c) this study. Neither study exhibits simple correlation without outlier rejection. A positive correlation between $Pb_c$ and $Pb^*$ may indicate presence of inclusions (mineral and/or melt), which were not removed during the chemical abrasion process.

## 5.2 Residual Pb loss in chemically abraded natural zircon:

The effect of Pb loss may be mitigated by removing decay damaged (metamict) portions of the zircon grains that have undergone open system behaviour, using the chemical abrasion procedure (Mattinson, 2005; Mezger and Krogstad, 1997). Subsequently, if all Pb loss domains were removed, the isotopic analysis should yield a concordant result for both the $^{206}$Pb-$^{238}$U and $^{207}$Pb-$^{235}$U decay series, respectively. However, the chemical abrasion may not have removed 100% of metamict portions and some domains with partial Pb loss may still be present in the grain (so-called residual Pb loss). This may be the case even if the most recent calibration of the chemical abrasion procedure (12hrs at 210°C, Widmann et al., 2019) is utilized. It should also be noted that chemical abrasion calibration may be affected by geochemical composition, parent isotope zonation, accessibility of damaged domains for the chemical agent and other effects. Previous work has shown that natural zircon reference materials (Temora and GJ-1) treated at 180°C and 210°C for 12hrs retains excess scatter in their U–Pb systematics, restricting repeatability confidence to ca. 0.1% of the absolute age, while synthetic solutions offer a repeatability at a precision of up to 0.01% in the same study (Schaltegger et al., 2021). It is however not clear if excess scatter in Temora and GJ-1 derives from inherited older components, magmatic processes or residual Pb loss for these materials (Schaltegger et al. 2021). When comparing the effects of different chemical abrasion temperatures, Huyskens et al. (2016) found that temperatures of 190°C and lower may yield incomplete removal of Pb loss domains, in agreement with later experiments (e.g. Widmann et al., 2019; Schaltegger et al., 2021). Such an effect of incomplete removal of metamict domains biased by Pb loss may possibly be detected through analytical discordance between the two decay schemes (Condon et al. 2024), provided that the analytical precision, especially of the $^{207}$Pb/$^{235}$U decay series, was sufficient, assuming that no inheritance is present (possible in the new dataset, not resolvable in Lindskog et al. 2017 dataset).

Between the previously published age estimates for the 'Likhall' bed by Lindskog et al. (2017) (467.50±0.28 Ma) and Liao et al. (2020) (465.18±0.17 Ma), there is a discrepancy of ca. 0.4% between the proposed U–Pb ages (Fig. 2). The two studies differ in their chemical abrasion protocols: Lindskog et al. (2017) utilised a 180°C and 12hrs procedure, whereas Liao et al. (2020) utilised a 190°C and 15hrs procedure. Both of these protocols diverge from the Widmann et al. (2019) parameters and we must consider the potential that these analyses included relict domains of Pb loss, that may bias the U–Pb dates towards younger dates for both studies. Curiously, for the first case, the lower T abrasion resulted in (on average) older interpreted zircon U–Pb data (Lindskog et al., 2017), an effect we will explore in the next section discussing blank corrections.

Both the Lindskog et al. (2017) and Liao et al. (2020) datasets show very little evidence for normal discordance, as analyses which are discordant are typically older than the interpreted U–Pb age, suggesting inheritance rather than Pb loss. Negative discordance can be masked if the analytical precision is insufficient, in particular when the measured $^{207}$Pb intensity is low. In our new data we observe two discordant analyses that are by ~2–4 Ma (Fig. 1) younger than the previous age estimates (Lindskog et al., 2017; Liao et al., 2020). We therefore infer that the data of Liao et al. (2020) and Lindskog et al. (2017) were

affected by Pb loss domains and/or relict inclusions that were not penetrated during chemical abrasion (expressed through elevated $Pb_c$ and/or younger dates), which may also not be excluded for our new dataset with absolute certainty. Relict inclusions have an unknown amount of $Pb_c$ and its composition cannot be assessed by ID-TIMS data. This could result in erroneous blank composition corrections, which may result in older U–Pb dates. Therefore, we base our discussion and
interpretation on those data that are the least affected by Pb blank correction.

### 5.3 Lead blank isotopic composition correction effects on the spread of zircon U–Pb dates:

One of the fundamental assumptions in zircon U–Pb geochronology is that zircon does not incorporate Pb during crystallisation and therefore does not contain initial (which can be monitored through analysis of $^{204}Pb$) (Watson et al., 1997). Therefore, all zircon U–Pb analyses can be corrected for the presence of common Pb through measurement of $^{204}Pb$ during data acquisition,
assuming that all $Pb_c$ is derived from the procedural blank, by adopting the mean isotopic composition from repeat analysis of procedural blank measurements. The uncertainty of this mean blank isotopic composition is propagated into the U–Pb date calculation (Schmitz and Schoene, 2007). An accurate and precise correction of blank Pb thus results in a more precise and accurate U–Pb dates. Conversely, inaccurate blank corrections may result in apparently too old or young U–Pb dates, if the $Pb*/Pb_c$ is low (Schaltegger et al., 2021).
In the Lindskog et al. (2017) dataset, a significant proportion of the analytical uncertainty is controlled by the Pb blank composition correction, as $Pb*/Pb_c$ ratios range from 2 to 22, while the Liao et al. (2020) dataset ranges from 2 to 30 (plus one high value at 67). In our new dataset, we observe $Pb*/Pb_c$ from 11 to 78. When we compare the $^{206}Pb/^{238}U$ zircon U–Pb dates with their $Pb*/Pb_c$ (Fig. 3), we observe that the Lindskog et al. (2017) data exhibit a slightly negative correlation, whereas the Liao et al. (2020) are scattered. However, we observe in the Liao et al. (2020) data that the youngest analyses are consistently
associated with lower $Pb*/Pb_c$, with the exception of the analysis with the highest $Pb*/Pb_c$ which is also relatively young (Fig. 3). In our newly acquired dataset, correlation between $^{206}Pb/^{238}U$ zircon U–Pb date with its $Pb*/Pb_c$ is absent, suggesting that blank correction does not introduce significant bias (Fig. 3).

The strong correlation between $Pb*$ and $Pb_c$ ($R^2$ of 0.52 after rejecting two outliers; $Pb*$ max = 55.33 pg and $Pb_c$ max = 5.24 pg) in the Lindskog et al. (2017) dataset is concerning (Fig. 4), as it implies that the analysed zircons contained inclusions that
were not removed during the chemical abrasion procedure. Evidence for potential inclusions (before chemical abrasion) is provided by imaging of zircon crystals analysed by Lindskog et al. (2017) and Liao et al. (2020), matching our observations during mineral selection. If we assume that larger zircons contain more $Pb*$ and a larger volume of $Pb_c$-bearing inclusions, this would explain why the Lindskog et al. (2017) dataset exhibits a negative correlation between $Pb*/Pb_c$ and $^{206}Pb/^{238}U$ dates, as they are more strongly affected by an inaccurate blank correction. The slope of the correlation between $Pb*/Pb_c$ and $^{206}Pb/^{238}U$
dates is controlled by the difference between the "true" $Pb_c$ composition and the assigned $Pb_c$ composition for blank correction. In the Liao et al. (2020) dataset, correlation between $Pb*$ and $Pb_c$ is absent, but measured $Pb*/Pb_c$ are comparable to those of Lindskog et al. (2017; Fig. 4). Our new dataset does not show any correlation between $Pb*$ and $Pb_c$ ($Pb*$ max = 59.80 pg and $Pb_c$ max = 1.09 pg; Fig. 4) nor between $Pb*/Pb_c$ and $^{206}Pb/^{238}U$ date (Fig. 3).

Some of the low Pb* zircons in our new data also have elevated blanks, causing scatter in the low Pb* analyses systematics,
whereas all the high Pb* analyses associate with slightly higher blanks. We therefore consider it likely that the $Pb_c$ is primarily
controlled by un-resolved inclusion (transparent, mineral and/or melt). These observations are in line with the observations by
McKanna et al. (2023a) and the Lindskog et al. (2017) data (Fig 4a), that chemical abrasion at high temperatures is necessary
to effectively remove inclusions that are deeply seated within the zircon crystal. Factors such as location of inclusions within
or outside of high U, damaged domains may impact the efficiency of the chemical abrasion procedure to remove inclusions.



## 5.4 The impact of the interpretation strategy on U–Pb zircon ages

Correlation between U–Pb dates and $Pb^*/Pb_c$ has implications for interpreting the absolute age of the 'Likhall' bed. Several
different interpretation strategies exist, such as i) weighted mean of a subset of data (e.g. Lindskog et al. 2017), ii) youngest
cluster of overlapping analyses at $2\sigma$ (e.g. Liao et al., 2020), iii) considering the entire range of concordant zircon U–Pb
analyses as autocrystic growth within the magma chamber (Samperton et al., 2015), iv) considering the youngest concordant
analysis as best proxy for the timing of eruption and v) applying a stochastic (Bayesian) sampling approach (Keller et al., 2018,
Traylor et al. 2021). We discuss in the following the impact of interpretation scenarios i–iv on the suggested age for the
'Likhall' bed (v is beyond the scope of this study, but we report model results in Table S2, which we find to be too heavily
dependent on the input data itself.). Analytical and mineralogical effects such as using variable or inaccurate blank isotopic
composition and the presence of $Pb_c$-rich inclusions, together with the presence of residual Pb loss and minor inheritance of
old radiogenic Pb (potentially from inherited cores), add to any of the further discussed discrepancies.



### 5.4.1 i) Subset interpretation

Lindskog et al. (2017) preferred a data interpretation based on the statistically most robust weighted mean age, representing
the largest number of statistically valid analyses, which results in 467.50±0.28 Ma (MSWD = 1.4, n = 9, published value
rejecting two younger and six older, concordant analyses). The underlying assumption is that some un-resolved Pb loss may
affect the youngest analyses and that inheritance may explain the oldest range of zircons analysed (Samperton et al., 2015).
Applying the same strategy to the Liao et al. (2020) dataset, we obtain 466.34±0.20 Ma (MSWD = 0.99, n = 12), rejecting five
younger and three older analyses. For our new dataset, the result would be 466.875±0.074 Ma (MSWD = 0.87, n = 9), rejecting
3 discordant analyses, 6 younger analyses and 5 older analyses. The maximum difference between the absolute ages amounts
to 0.25%, which is better than the difference of 0.4% of the previously published values. The lowest discrepancy is achieved
between Liao et al. (2020) and our dataset, with a difference of 0.11%.

285

290

### 5.4.2 ii) Youngest cluster interpretation

Liao et al. (2020) preferred an interpretation based on the youngest, statistically valid age cluster, which resulted in 465.18±0.17 Ma (MSWD = 1.3, n = 5, published value). The underlying assumption is that all Pb loss is effectively removed and that inheritance or protracted growth in the magma chamber is responsible for the older analyses. For the Lindskog et al. (2017) data, this approach results in 466.96±0.30 Ma (MSWD = 1.5, n = 6). For our new dataset, the corresponding result is 466.43±0.11 Ma (MSWD = 5, n = 5). The resulting maximum spread is 0.38%, close to the published discrepancy of 0.4%. The lowest discrepancy is achieved between Lindskog et al. (2017) and our dataset, with a difference of 0.11%.

### 5.4.3 iii) range of concordant zircon U–Pb dates

The duration of zircon crystallisation in a magma chamber is of interest for studies involving magma chamber dynamics and crustal evolution and we here compare the $\Delta t$ defined as spread between youngest and oldest (concordant) zircon U–Pb analysis. The underlying assumption is that all concordant analyses reflect growth from the same magma chamber (Samperton et al., 2015). The $\Delta t$ for the Lindskog dataset is 5.35 Ma, Liao et al. (2020) is 3.8 Ma (rejecting two outliers marked by Liao et al. (2020)) and our new dataset is 1.82 Ma (rejecting 3 young discordant data points) – results which differ significantly from each other. One possible explanation could be the unresolved inclusions and resultant low $Pb^*/Pb_c$, which can result in calculated too young or too old dates, augmenting natural spreads in zircon U–Pb dates. The maximum difference between the three datasets is 194%, suggesting that interpretation is problematic with respect to the duration of magma chamber activity.

### 5.4.4 iv) youngest concordant zircon U–Pb analysis

The youngest concordant zircon U–Pb analysis interpretation can be useful in volcanic samples. Here, the base assumption is that zircon continuously crystallises in the magma, and the youngest zircon represents the best proxy to the timing of eruption. In this regard, for Liao et al. (2020) the youngest concordant zircon U–Pb analysis is 465.07±0.17 Ma, for Lindskog et al., (2017) it is 466.03±1.06 Ma and for our new dataset it is 466.05±0.29 Ma. The maximum difference between the three datasets is 0.21%, better than the discrepancy of the published U–Pb dates. There is a near indistinguishable difference between the Lindskog et al. (2017) and our youngest zircon U–Pb analyses (0.004%).

## 5.5 Which strategy to choose?

It becomes clear from Fig. 2 that the three datasets were obtained under different analytical regimes in different labs. Among the differences is an improvement in analytical precision (increase of $Pb^*/Pb_c$ through blank reduction). This will help to narrow down the duration of zircon growth in the magma system prior to eruption and leads to more reliable identification of autocrystic vs. antecrystic zircon. Identifying Pb loss remains challenging in the older datasets, which utilized partial dissolution procedures at lower temperature and/or of shorter duration. Furthermore, low $Pb^*/Pb_c$ obscure discordance between the two decay series, due to elevated $^{207}Pb/^{235}U$ uncertainty. The largest difference in age, however, is caused by the varying

approaches to data interpretation. If the same strategy is chosen, the discrepancy between the proposed U–Pb age in the two previously published studies and the present one significantly decreases.

We can conclude that Pb*/Pb$_c$ is of fundamental importance for the 'Likhall' zircon datasets. If we, for example, consider
only analyses that are characterised by Pb*/Pb$_c$ >50 – i.e., a value where zircon dates are barely affected by blank correction – we would have to reject the entire Lindskog et al. (2017) dataset and all but one analysis from the Liao et al. (2020) study, and it leaves only 8 out of 22 analyses from our new dataset, of which only 3 form a cluster of youngest, overlapping grains. Adopting this reduced dataset for interpretation yields the following results: i) the largest number of overlapping U–Pb dates forms a weighted mean of 466.76±0.12 Ma, ii) the youngest cluster is 466.37±0.14 Ma, iii) the Δt reduces to 780 kyrs, which
is more consistent with predictions of thermal models for magma chambers (e.g., Caricchi et al., 2016; Weber et al., 2020), and iv) the youngest concordant zircon analysis remains unchanged at 466.05±0.29 Ma. The maximum difference between interpretations i), ii) and iv) is 0.15%, close to desired reproducibility value of 0.1% for natural reference materials (Schaltegger et al., 2021). More rigorous filtering based on Pb*/Pb$_c$ thus yields a more coherent dataset between the three studies, however, at low n-value.

Therefore, we propose that the weighted mean age of the youngest cluster of three high-Pb*/Pb$_c$ analyses at 466.37±0.14/0.18/0.53 Ma (analytical/ +tracer/ +decay constant uncertainty; MSWD=1.9, n=3) has the highest probability for being an accurate age estimate for the 'Likhall' bed.

## 5.6 Implications for the Ordovician timescale and the absolute timing of events

Our new age for the 'Likhall' bed provides a carefully considered and well constrained tie point in the overall Ordovician
timescale and event-stratigraphic framework. In the Geological Time Scale 2020 (GTS2020), the level corresponding to 'Likhall', i.e., the basal *Yangtzeplacognathus crassus* conodont Zone, is placed at c. 469 Ma (Goldman et al., 2020, figure 20.3). This is well outside our new age estimate of 466.37±0.14/0.18/0.53 Ma (and also those of Lindskog et al., 2017 and Liao et al., 2020), and thus warrants adjustment of the Ordovician timescale – especially given that the range of the *Y. crassus* Zone in the GTS2020 scheme does not even overlap with our age estimate. The revised age for 'Likhall' further suggests that
the timing of the L-chondrite breakup event in space, as interpreted based on cosmic-ray exposure age data from 'fossil' meteorites, should be placed at c. 467 Ma (cf. Korochantseva et al. 2007; Heck et al. 2008; Lindskog et al. 2017; Liao et al. 2020; and references therein).

We refrain from extending our data interpretations until reliable U-Pb dates from both older and younger strata are available. Nonetheless, it is apparent from the relative timing in the rock record that the purported meteorite bombardment significantly
postdates the onset of climate change and biodiversity spurts in the Middle Ordovician (e.g., Lindskog et al. 2017, 2023; Rasmussen et al. 2021). These stratigraphic offsets are unlikely to be bridged by numerical dating, but continued efforts to refine the Ordovician timescale will provide essential temporal context to the long-term development of the geobiosphere.

**6 Conclusions**

**1)** Our new high-precision $^{206}Pb/^{238}U$ data set of the 'Likhall' zircon population represents an analytical improvement over previously published results: (a) our new data show higher $Pb^*/Pb_c$ and therefore are less affected by the choice of isotopic composition used for blank correction, (b) higher analytical precision allows for identification of normal discordance in young analyses, (c) the presence of residual Pb loss despite application of the currently considered optimal parameters for chemical abrasion (210°C for 12hrs; Widmann et al., 2019) suggests that previous data acquired at lower chemical abrasion temperatures were (likely more) affected by Pb loss domains. The elevated $Pb_c$ in the former studies is suggested to be due to incomplete removal of $Pb_c$-rich inclusions during chemical abrasion.

**2)** The choice of the data interpretation strategy is the main reason for the discrepancies between datasets from the three different laboratories (Lindskog et al., 2017; Liao et al., 2020; and the present study), particularly in cases where elevated $Pb_c$ causes decreased analytical precision. Further reduction of the reproducibility between these studies is caused by the use of different tracer solutions and their respective calibrations.

**3)** The difference in interpreted age drastically decreases when only data subsets with high $Pb^*/Pb_c$ are considered. For the data sets considered here, this implies an empirical $Pb^*/Pb_c$ threshold of 50. For analyses with $Pb^*/Pb_c$ >50, the weighted mean (youngest cluster) U–Pb zircon age for the 'Likhall' bed is 466.37±0.14/0.18/0.53 (analytical/+tracer/+decay constant uncertainties), suggesting that previously published U–Pb ages are inaccurate.

**4)** The efficient removal of Pb loss domains and micro inclusions is of paramount importance for achieving an accurate U–Pb date. Considering the presence of discordant, young analyses in our Ordovician-age dataset despite chemical abrasion conditions that are considered to be optimal, there is incentive to further develop the chemical abrasion procedure.

**5)** Considering biostratigraphic aspects, compared to GTS2020, the revised age of the 'Likhall' bed necessitates significant internal adjustment of the Ordovician timescale.

**Author Contributions**

Conceptualization: André N. Paul, Anders Lindskog, Urs Schaltegger

Data Curation & Formal Analysis: André N. Paul

Funding Acquisition: Anders Lindskog and Urs Schaltegger

Investigation: André N. Paul, Anders Lindskog, Urs Schaltegger

Visualization: André N. Paul and Anders Lindskog

Writing – Original Draft Preparation: André N. Paul

Writing – Review & Editing: André N. Paul, Anders Lindskog, Urs Schaltegger

## Declaration of Competing Interests

The authors declare that they have no conflict of interest.

## Acknowledgments

A.L. acknowledges funding from the Birgit and Hellmuth Hertz' Foundation and the Royal Physiographic Society of Lund. This study benefitted from SNFNS grant CRSII5_180253 awarded to U.S. We thank three anonymous reviewers for their constructive comments that helped to improve the manuscript. We thank associate editor Sandra Kamo and editor Klaus Mezger for handling the manuscript and providing constructive comments that improved the quality of the manuscript.

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
