# Peer review of "Short communication: Resolving the discrepancy between U-Pb age estimates for the 'Likhall' bed, a key level in the Ordovician timescale"

_EGUsphere, 2023_

## Referee Comment (RC2)

The paper presents a third U-Pb zircon data set for zircons recovered from the Ordovician Likhall carbonate horizon in Sweden. The data appears to be superior to previous efforts in that they have successfully increased the ratio of radiogenic to common Pb in a subset of zircons that they use to determine their preferred age. The data appear to be of high quality and the new age an improvement that will have impact on the Ordovician timescale and a debate concerning a possible link between the Great Ordovician Biodiversification Event (GOBE) and the break up and arrival of an L-chondrite asteroid. On these merits alone, the paper should be considered worth publishing.

However, the paper gets seriously sidetracked in its effort to assess the two previously published U-Pb zircon studies of this same horizon (same samples in at least one other study as the author Lindskog provided the zircons to both). In this endevour, they provide a confusing (and in places factually incorrect) discussion of why they believe the different studies have delivered different ages. They ultimately suggest that the other ages are inaccurate but then go ahead anyways and try to assess the different methods of treating a range of ages from a volcanic horizon that only makes sense if all the individual zircon ages are assumed to be accurate.

While the paper expends much effort in the comparison of different data sets and recommendations for the chemical abrasion method, there is only 8 lines of discussion (Section 5.6) of the implication of their urevised age on timescale issues and no mention that I found of the implication for the debate on whether the Ordovician L-chondrite break up had anything to do with the GOBE.

I would recommend that the authors be encouraged to make major revisions to the paper, toning down the detailed assessment of the two previous studies. They can demonstrate by way of the Pbr/Pbc that their data (and uncertainties) are better than previous efforts and then focus on the implications.

I have annotated the manuscript extensively with comments and provide the following more general points keyed to different sections of the discussion that I feel are important.

*5.1 Radiogenic Pb/common Pb ratio (Pb\*/Pbc) as a selection criterion*
In this section, the authors suggest that the zircons analysed in Lindskog et al had inclusions that were not removed in the partial dissolution step of the chemical abrasion method. While this may be true, they have missed or decided to ignore the point that this study assumed that all non-radiogenic Pb was blank (or modern Pb that was not removed from the zircon before digestion). If it is initial common Pb it would have an Ordovician Pb composition that would have had lower 7/4 and 6/4 ratios resulting in older, not younger, ages. They never recognize this problem and seem to consider common Pb, blank Pb and initial Pb as one and the same.

*5.2 Residual Pb-loss in chemically abraded natural zircon*

After concluding in section 5.1 that the Lindskog et al data was troubled by low Pbr/Pbc, they explore here a different tack in that perhaps there were residual domains that had lost Pb that were not removed by the lower temperature etching step employed in this study. But this would have the opposite effect of producing younger ages rather than the older ages they are trying to explain away. L205 states that they conclude that both previous studies "*were affected by Pb-loss domains and/or relict inclusions that were not penetrated during*

*chemical abrasion.”* This may be true for the Liao et al study that recommended a younger age but not the Lindskog et al data as suggested.

*5.3 Lead blank isotopic composition correction effects on the spread of zircon U–Pb dates*

Here correlations between Pbr and Pbc are considered in terms of their absolute abundances without considering the difference between Pbc as Pb blank or Pb initial and that assuming one or the other would have different implications for the discussion. Again, they fail to discuss the Lindskog et al paper having assigned all non-radiogenic Pb as modern terrestrial Pb and that their interpretation of Pb-bearing inclusions of presumed Ordovician age would have lower 6/4 and 7/4 ratios, producing older calculated ages in both decay schemes. This makes the different in ages between the two studies worse rather than explaining it away.

*5.4 The impact of the interpretation strategy on U–Pb zircon ages*

This section runs through the different established strategies employed to make a sensible interpretation of a spread of U-Pb zircons commonly found in volcanic ash layers. A combination of issues may be in play here to cause the spread including residence time of zircons (related to magma chamber timescales) and/or detritial grains from an earlier eruptions but issues that have geological explanations. These strategies assume that all the individual ages of zircons in the range are accurate for their respective crystallization and closure. But previous sections have concluded that the 2 previous studies and especially that of Lindskog et al are not reliable ages due to their high Pbc. If they have concluded that at least the Lindskog et al ages are inaccurate then they should not be considered further in this section. Only the zircons with their cut off of >50 for Pbr/Pbc should be considered here.

*5.6 Implications for the Ordovician timescale and the absolute timing of events*

This section is eight lines long without discussing the implications for the debate mentioned in the introduction regarding the connection between the Ordovician break up and arrival of the L-chondrite and the GOBE or any implications for the new age on the Ordovician timescale save that it needs to be modified. This should be the most important part of the paper else why bother to get a refined age for this horizon in the first place. As presented, it seems the main justification for this study was to show that that you need higher temperatures and longer times for the chemical abrasion method.

[revised manuscript text omitted]

---

## Referee Comment (RC3)

Thanks to the editor and authors for the opportunity to review this interesting paper. This study revisits a previously dated zircon-bearing carbonate bed, the Likhall bed, which has important implications for the Ordovician time scale but has yielded inconsistent dates between studies. The authors attribute these differences to both

- methodological issues (mainly incomplete chemical abrasion by previous studies, as well systematic differences in tracer calibration between studies using an ET spike and a study using an in-house spike) and
- interpretive differences relating to how one determines a single age for a bed that contains multiple dated zircon grains.

The authors grapple with these problems and put forward a new weighted mean age estimate for the bed. The study is well-written, concise, and clear. I support publication of the study and make the following (hopefully constructive) suggestions.

This bed was previously dated by two studies, Lindskog et al. 2017 and Liao et al. 2020, and ages from all three studies are shown in Figure 2.

- This figure shows all the ages with their X uncertainty, but I believe they should all be shown with their Y uncertainty (including tracer calibration uncertainty), as the Liao data were determined using an in-house spike and the Lindskog/Paul data were generated using an ET spike. If the authors don't wish to or can't do this, then I suggest caveating this in the caption.

Line 95: Can the authors provide a citation supporting the assumption of a magma initial Th/U ratio of 3.5 +/- 1?

Figure 1:

- I suggest recoloring this figure so that the cut-off value of 50 is at an inflection point in the colorscale and/or indicate on the colorscale where 50 is. It is hard to look at this and immediately understand that some of the green grains that overlap with the concordia curve are actually below 50 and should be left out.
- It's also somewhat confusing that the grains "selected" for analysis here are not the same as the grains selected in Table S1, column "interpretation strategy i."
  - I suggest annotating this column in Table S1 to indicate the three grains that are selected for analysis using the more restrictive Pb*/Pbc screen.
  - I suggest adjusting the figure so that all grains that are "selected" using "interpretation strategy i" have a black dashed ellipse and those that are "selected" using the more strict Pb*/Pbc screen have a black ellipse (as they do now).
- Isn't a part of the selection criteria also that the ages are within the youngest cluster (line 305)? This should be stated in the caption.

Line 125: I think the geological setting should be given greater priority within the manuscript and encourage the authors to include a figure showing the field context of the bed sampled. I understand that this has been described elsewhere, but such a figure would increase the value of this manuscript to the reader. Similarly, images of the zircon grains have been previously published elsewhere, but including some images in this manuscript or in the supplement would be helpful.

The authors first consider differences in the chemical abrasion techniques used between studies, and argue that incomplete chemical abrasion of inclusions and metamict zones within zircons resulted in inaccurate ages for the previous studies. Previous studies used lower temperatures for their chemical abrasion procedures than the current study. The main data the authors use to support this idea is shown in Figure 4, which plots Pbc vs Pb* for each of the three studies.

- This is more of an aesthetic suggestion, but I suggest making all the plots the same size and aligning their left y-axis.
- I also think that plotting all the data with the same axes or on the same plot would help the authors make their point that their approach has resulted in much smaller Pbc measurements and presumably more precise and accurate dates.
- Because chemical abrasion is meant to remove zones of zircon affected by Pb loss, it's easy to follow this argument for the Liao data, which the authors argue is too young because of incomplete chemical abrasion, but harder to follow it for the Lindskog data, which is too old (and the authors discuss this). They suggest that the "too old" ages come from inaccurate and imprecise blank corrections.
  - Line 231: "If we assume that larger zircons contain more Pb* and a larger volume of Pbc-bearing inclusions…" I'm happy to agree that larger zircons would have more Pb* and Pbc, but I'm missing a step in the logic of the sentence here. I think it's that larger zircons with more Pbc would be more impacted by an erroneous blank correction, but perhaps it can be spelled out a bit more clearly.

The authors consider several different interpretation strategies. They identify five ways that one might determine the age of this bed:
1. Weighted mean of a subset of data
2. Youngest cluster of overlapping ages at 2-sigma
3. Use entire range of concordant zircon analyses as autocrystic growth in magma chamber
4. Use the youngest concordant grain as the best proxy for the timing of eruption
5. A Bayesian approach as suggested by Keller et al. 2018

- I would rephrase 5 to something like "A Bayesian approach such as those suggested by Keller et al. 2018 or Traylor et al. 2021."
  - The authors immediately state that #5 is outside the scope of this study, but I don't agree. The approach used by Keller et al. 2018 is relatively easy to implement and a publicly-available Jupiter Notebook provides help at https://github.com/brenhinkeller/Chron.jl?tab=readme-ov-file. I suggest that it be included and discussed; if the authors have a major disagreement with this approach, they should show why.
  - Traylor et al. refers to modifiedBChron, which would consider all analyses as part of a summed probability density distribution function and then use stratigraphic superposition in a Bayesian model to determine an age that is supposed to be a better representative of geological uncertainty. Using only the new data produced by this study, of course this can't be applied as we don't have the benefit of stratigraphic superposition, but if, as the authors state, there are other dated bentonites in the section with clear stratigraphic relationships, why not give it a try, and see how it compares? modifiedBChron is also fairly easy to implement. I

suspect the outcome will mostly highlight the importance of getting good age constraints elsewhere in the section, but this would be worth highlighting too.

The Bayesian approaches introduced by Traylor and Keller are exciting new developments that help us understand what the meaning of a dated ash bed means in stratigraphic context. This paper's ambition is to make the point that the Ordovician timescale must be reconsidered, and it certainly provides evidence in that direction, but at present, this paper misses an opportunity to grapple with these emerging approaches and their implications for the timescale.

The authors spend a lot of time discussing using Pb*/Pbc as a screening metric and note that it significantly improves accuracy while coming at a significant cost of greatly reducing the number of grains that are viable. Can they give non-zircon geochronologists some sense of how a strict requirement of Pb*/Pbc > 50, for example, would affect the universe of published CA-ID-TIMS zircon data? Would it knock out >50% of published zircon grains, as it does here? Given the expense and time-consuming nature of CA-ID-TIMS analysis, how do they recommend other workers grapple with this?

Line 299: "leaves only 8 of 22"—yes and worth stating explicitly that these 8 are not all within the youngest age cluster.

The authors describe some of the changes to the Ordovician timescale that their new age requires.
- It would be helpful for the authors to visualize these changes in a "before" and "after" figure.
- The authors also note that the timing of L-chondrite breakup should be revised to c. 467.1 Ma, but the reader isn't given enough context about why this date should be assigned to the breakup, (which isn't the age they give the Likhall bed).
- As the authors motivated this study by mentioning the controversial hypothesis linking meteorite bombardment with Ordovician biodiversification in the introduction, they should return to this hypothesis and discuss the alignment or un-alignment of these two events in light of their new results.

Finally, the references:
- Liao et al. 2020 is a key study for this paper and the source of much of the legacy data discussed, but it does not appear in the reference list. This must be addressed before final publication.
- I suggest reformatting the Schmitz et al. citation in the references to superscript 205, 235, 238, etc. Same thing for the von Quadt et al. citation and 10^13.
- On the topic of CA-ID-TIMS dating of cryptotephra from carbonates: The authors might be interested to know about a similar study focused on retrieving volcanic zircon from carbonate rock: Finzel and Rosenblume 2021, *Geology*. I think referencing this study would help make the authors' point that determining best interpretive practices for carbonate-derived crypotephra ashes is a question with applications beyond the single bed considered here.

---

## Author Response (AR1)

Response to Reviewer #1

This study presents a new data set U-Pb zircon dates for the Ordovician carborate-rich 'Likhall' zircon bed, as well as a critical assessment of two published U-Pb zircon data sets for the same zircon population. The quality of the new U-Pb data in this study is top-notch, but the manuscript needs improvement in two other aspects: mineralogical support, and attention to detail in interpretation and discussion. The detailed comments linked to the text are below.

Dear reviewer, we firstly would like to thank you for taking the time and effort to review and comment on the manuscript. Below, we are responding to your line specific comments and the outline the changes made to the manuscript.

Lines 43-44. The paper by Liao et al. (2020) is mentioned in the manuscript 33 times, but is omitted from the reference list. Please add it.

We have added the missing reference.

Lines 45-47. "Most importantly, chemical abrasion procedures also differ, being 180°C for 12hrs (Lindskog et al., 2017) and 190°C for 15hrs (Liao et al., 2020). ". Conditions of chemical abrasion include not only leaching time and temperature, but also annealing time and temperature. The latter can be just as important, and should be discussed.

We have added the mentioned details. The discussion of these parameters follows immediately after, as in the original submission. This concerns lines 45-51.

Lines 57-60. "The 206Pb/238U age of the natural reference zircon material Temora reported by Liao et al. (2020) from the mixed TIMS–MC-ICP-MS analysis (417.19±0.15 Ma) is in agreement with the most recent estimates of 417.310 ± 0.074 Ma (von Quadt et al., 2016) and 417.353±0.052 Ma (Schaltegger et al., 2021).". While this justification is reasonable, it is compromised by the fact that Temora is a natural material, and has been demonstrated to be heterogeneous. The role of using different spikes can be (and must be) evaluated more precisely and reliably by direct comparson of calibrations of the two spikes. Liao et al. (2020) used the spike that was described by Huyskens et al. (2016). That spike was calibrated against two gravimetric U-Pb solutions that were also used by Condon et al. (2015) to calibrate their ET2535 spike. The difference in calibration therefore includes only the calibration procedure itself (about 0.05% or less combined calibration uncertainties from both studies), but not the uncertainty in composition of gravimetric solutions.

We added more detail through reported ET100 data of the Liao et al. study lab, our data, and the suggested reference value of the ET100 solution. Lines 60-62.

Lines 72-74. "Single grains of 'Likhall' zircon crystals free of visible inclusions and cracks were hand-picked under a binocular microscope at a magnification of ×20 to ×40 from the same petri dish as the material analyzed by Lindskog et al. (2017). The size of individual fragments was variable, with length ranging from ~50μm to ~300μm.". CL/BSE images, or at least optical images, of the analysed zircons would be useful.

We would agree that CL/BSE and optical images can be useful, but considering that two previous studies have provided such without any benefit to the U-Pb age interpretation we see no additional value of these in this study. In the ID-TIMS community recommendations (Condon et al. 2024, GSA), these are also classified as optional, and eventually we have opted against them, considering from the same material are already images published.

Line 96. "(206Pb/238U date: 100.173 ± 0.003 Ma; Schaltegger et al., 2021)". This is unfortunate that the primary gravimetric data for the ET age reference solutions are still not

published, and we have to use the mean results from previous studies as a reference value. We agree that the responsible scientists should publish the primary gravimetric data for the ET reference solutions.

Line 97. "n = 32/40". This is a massive amount of rejection. What mean value and uncertainty do you get if you include all data?
The mean value and uncertainty without any data rejection is 100.1639±0.0043 Ma, compared to the preferred reported age of 100.1678±0.0046 Ma. We would like to point out that in Line 97-98 we mention that one batch of the ET 100 is rejected, for an anomalously young average age. The argument of rejection of this batch from overall consideration stems from the process and interlab comparison observations in Schaltegger et al. (2021). The ET 100 solutions are routinely prepared in batches, which are split into aliquots after processing (1 batch yielding 5-8 aliquots), as reported and recommended in Schaltegger et al. (2021) (process described in detail in that study). In Schaltegger et al. (2021) there is reported data and discussion of similar observations from different labs, an issue that remains to be evaluated and solved by the Earth time community. We will ensure that the full ET 100 data is available in the supplementary repository or similar.

Line 124. "4 Geological Setting". Why is this section placed after "results"? It would be more appropriate to put it in the more conventional place after Introduction and before Methods.
The Geological setting is now in position "3".

Lines 152-153. "Subsequently, we only use high Pb*/Pbc (>50) analyses for our age interpretation and will also apply this strategy to the previously published datasets.". Setting the cut-off of Pb*/Pbc at 50 would be reasonable for evaluation of the data of this study only. But it excludes almost all data from the two previous studies. I suggest to repeat evaluation at two cut-off levels, e.g., 15 and 50 (i.e., <15, 15-50, and >50 bands). The lower level would include sufficient number of data points from the published studies and yield a more objective and informative comparison.
We have calculated the suggested bands and briefly discuss them in the manuscript. Weighted mean ages of all three datasets of the band 15-50 overlap within uncertainty but are slightly older (0.1%) than our preferred date from interpretation of zircon with Pb*/Pbc >50. At <15 the weighted mean ages scatter significantly. See lines 184-196.

Lines 184-185. "This may be the case even if the optimal calibration of the chemical abrasion procedure (12hrs at 210°C, Widmann et al., 2019) is utilized.". These conditions were established as optimal by analysis of just one zircon reference material. How can we be sure that they are optimal for any zircon in general?
We rephrased, including the possibility that compositional and other effects can influence chemical abrasion results. Lines 226-227.

Lines 191-193. "Such an effect of incomplete removal of metamict domains biased by Pb-loss may possibly be detected through analytical discordance between the two decay schemes, provided that the analytical precision, especially of the 207Pb/235U decay series, was sufficient.". This reasoning is valid only in the absence of inherited material.
We rephrased and state that this assumes absence of inherited material. Lines 234-235.

Lines 205-206. "We therefore infer that the data of Liao et al. (2020) and Lindskog et al. (2017) were affected by Pb-loss domains and/or relict inclusions that were not penetrated during chemical abrasion.". The part of this statement about inclusions equally applies to the current study, because there is no certainty in whether the chemical abrasion procedure of

Widnann et al. (2019), or any specific chemical abrasion protocol, guarantees removal of inclusions of other minerals (which can potentially include some minerals that are even more difficult to dissolve than zircon, e.g., cassiterite).

The reviewer is correct in pointing out that any zircon may exhibit relict Pb-loss after chemical abrasion. We already present evidence of Pb-loss in the 'Likhall" zircons in Fig. 1 of the original submission. Mentioned in line 204-205 is the observation that at least 2 analyses of our new dataset exhibit Pb-loss, which demonstrates that the Widmann procedure did not remove all Pb-loss in this study. We rephrased to clarify in line 249.

Line 218. "(Schaltegger et al., 2021).". The reference to Gaynor et al. (2022) Chem. Geol., who discussed this topic in detail, would be more appropriate here.

The reviewer is incorrect here. The study by Gaynor et al., 2022 discuss Pb*/Pbc and its effect on the total uncertainty. The study by Gaynor et al. (2022) does not however model the effect of changing blank isotopic correction (IC) on the absolute dates. Since we refer to the effect of changing the blank IC effect on the absolute date, we continue to refer to Schaltegger et al. (2021).

Lines 229-231. "Evidence for potential inclusions (before chemical abrasion) is provided by imaging of zircon crystals analysed by Lindskog et al. (2017) and Liao et al. (2020), matching our observations during mineral selection.". More details are needed here. What evidence? What inclusions (composition, size, abundance etc.)? What methods were used to study these inclusions?

We are explaining in more detail the nature of our observations, which are primarily observations made and noted during multiple picking sessions for the annealing, chemical abrasion, washing and dissolution steps. During the picking process, zircons with visible cavities and inclusions (these may be melt or mineral inclusions) are identified. While we take great care in examining zircons selected for analysis under the microscope, we cannot exclude with absolute certainty that sub-visible inclusions are present. Lines 78-79.

Lines 240-242. "These observations are in line with the observations by McKanna et al. (2023a) and the Lindskog et al. (2017) data (Fig 4a), that chemical abrasion at high temperatures is necessary to effectively remove inclusions that are deeply seated within the zircon crystal.". This does not make much sense. If the inclusions were embedded in zircon during crystallization, and are not confined to high-U growth zones, then they are hosted by crystalline zircon material. They would be released only during complete zircon dissolution. No chemical abrasion would help there.

We rephrased to clarify that the dependency of inclusion accessibility depends on structural features and not primarily on time and temperature. Line 287-288

Lines 250-251. "(v is beyond the scope of this study).". Why? This is most unfortunate. This could be one of the most interesting parts of this study. And it is not too late to add this comparison during revisions.

In our opinion, computing a model age based on data that may be flawed in some form is not advisable. In the Liao et al. dataset we'd assume presence of Pb-loss, in Lindskog et al. higher dependence on initial/blank Pb IC and in our new dataset Pb-loss and inheritance. These variables make it too difficult to make a good assessment in our opinion. Since reviewer 3 also requested the modelling to be made, they are made available in the appendix for documentation primarily.

Lines 251-252. "… and the presence of Pbc-rich inclusions". The presence of inclusions is not an "analytical effect". It is a mineralogical / geochemical feature.
We are clarifying that "analytical and mineralogical effects…" are considered.

Lines 255-256. "Lindskog et al. (2017) preferred a data interpretation based on the statistically most robust weighted mean age, representing the largest number of statistically valid analyses". Please briefly explain the underlying geochemical assumptions of this, and every other approach to data evaluation. Without that, the comparison of these approaches looks arbitrary.
We are not sure what the reviewer means with "geochemical" assumptions, however we are happy to expand the sections by briefly mentioning that:

i) The underlying assumption is that some un-resolved Pb-loss may affect the youngest analyses and that inheritance may explain the oldest range of zircons analysed (Samperton et al., 2015).

ii) The underlying assumption is that all Pb-loss is effectively removed and that inheritance or protracted growth in the magma chamber is responsible for the older analyses.

iii) The underlying assumption is that all concordant analyses reflect growth from the same magma chamber (Samperton et al., 2015).

iv) The underlying assumption is that all zircon analyses except the youngest is affected by inheritance or growth during protracted magma chamber activity (Schaltegger et al., 2009).

These changes are made in subsection 5.4.1 to 5.4.4

Line 278. "The youngest concordant zircon U–Pb analysis interpretation can be useful in volcanic samples.". Any interpretation of an age spectra based on a single data point is highly suspicious, especially since there may be still poorly understood factors that can make some grains particularly prone to Pb* loss (I suspect deformation is one of these factors).
As with any age interpretation, there is either statistical, geological or philosophical factors that are weighed. We here simply test previously published and executed interpretation strategies and explore their relative differences, without making any recommendation, thus don't see any need for changes.

Lines 291-293. "The two datasets produced with the EARTHTIME isotopic tracer ET2535 show better comparability despite divergent chemical abrasion procedures, pointing to a systematic effect of different tracer calibration as well.". This statement is unfounded. To compare tracer calibrations, you must look at tracer calibrations directly, not at some proxies. And the calibration data for both tracers are published (Condon et al. 2015, and Huyskens et al. 2016, respectively). Both calibrations were done against the same mixed gravimetric reference solutions. So if there is indeed a discrepancy between these spikes, it does not exceed the uncertainty of the calibration process itself (low 0.0x% values).
The statement has been removed from the manuscript.

Line 296. "We can conclude that Pb*/Pbc is of fundamental importance for the 'Likhall' zircon datasets.". This is the main conclusion of this study: to date reliably, we must get the ratio of radiogenic Pb to common Pb as high as possible (although this is not a novel idea). Furthermore, it is likely to be the only thing that really matters here. You have to admit that the evaluation of the roles of all other factors is tentative at best, and say so directly.

Neglecting the potential impact of interpretation strategy is an erroneous assessment by the reviewer in our opinion. Reducing the focus on Pb*/Pbc is a rather crude oversimplification. We clearly demonstrate that discrepancy between the interpretation strategies has amplified the difference in the published U-Pb dates when comparing the Liao et al.(2021) and Lindskog et al. (2017) studies.

Line 322. "(c) our new data show higher Pb*/Pbc". This point is the most important, and you should emphasize it, maybe by bringing it to the front. Higher precision is largely a result of lower common Pb correction. Whether the difference in chemical abrasion protocols actually influenced the data cannot be reliably deduced from the existing data sets.
We have moved point c) to a) in the revised manuscript.

Response to Reviewer #2

The paper presents a third U-Pb zircon data set for zircons recovered from the Ordovician Likhall carbonate horizon in Sweden. The data appears to be superior to previous efforts in that they have successfully increased the ratio of radiogenic to common Pb in a subset of zircons that they use to determine their preferred age. The data appear to be of high quality and the new age an improvement that will have impact on the Ordovician timescale and a debate concerning a possible link between the Great Ordovician Biodiversification Event (GOBE) and the break up and arrival of an L-chondrite asteroid. On these merits alone, the paper should be considered worth publishing.

However, the paper gets seriously sidetracked in its effort to assess the two previously published U-Pb zircon studies of this same horizon (same samples in at least one other study as the author Lindskog provided the zircons to both). In this endevour, they provide a confusing (and in places factually incorrect) discussion of why they believe the different studies have delivered different ages. They ultimately suggest that the other ages are inaccurate but then go ahead anyways and try to assess the different methods of treating a range of ages from a volcanic horizon that only makes sense if all the individual zircon ages are assumed to be accurate.

While the paper expends much effort in the comparison of different data sets and recommendations for the chemical abrasion method, there is only 8 lines of discussion (Section 5.6) of the implication of their urevised age on timescale issues and no mention that I found of the implication for the debate on whether the Ordovician L-chondrite break up had anything to do with the GOBE.

I would recommend that the authors be encouraged to make major revisions to the paper, toning down the detailed assessment of the two previous studies. They can demonstrate by way of the Pbr/Pbc that their data (and uncertainties) are better than previous efforts and then focus on the implications.

I have annotated the manuscript extensively with comments and provide the following more general points keyed to different sections of the discussion that I feel are important.

Dear reviewer, thank you for taking the time and effort to assess this manuscript submission. Below we respond to your general comments, while your annotations will be considered directly when producing a revised manuscript. We have opted to focus primarily on the

multiple factors that can sway a U-Pb age interpretation, to make the most confident interpretation of available data. We are happy to expand the section 5.6 a bit more, but the primary focus of our study lies on showcasing the sources of discrepancy between analyses of practically identical sample materials.

5.1 Radiogenic Pb/common Pb ratio (Pb*/Pbc) as a selection criterion

In this section, the authors suggest that the zircons analysed in Lindskog et al had inclusions that were not removed in the partial dissolution step of the chemical abrasion method. While this may be true, they have missed or decided to ignore the point that this study assumed that all non-radiogenic Pb was blank (or modern Pb that was not removed from the zircon before digestion). If it is initial common Pb it would have an Ordovician Pb composition that would have had lower 7/4 and 6/4 ratios resulting in older, not younger, ages. They never recognize this problem and seem to consider common Pb, blank Pb and initial Pb as one and the same. We agree that changing the IC of the blank correction will drive the Lindskog et al. (2017) data to either older or younger dates, subject to choosing a Pb blank IC. Given the radiogenic assignment of the Pb blank IC in Lindskog et al. (2017) (6/4 = 18.5), choosing a 500 Ma S&K value will make the Lindskog et al. (2017) data older. We can however, not assess if the original Pb blank IC is over- or underestimated in the Lindskog et al. (2017) study. Recognizing the correlation between U-Pb date and Pb*/Pbc, we interpret it as an artefact of inaccurate Pb blank correction. As the reviewer mentions in the annotated manuscript, we will explain more the effect on the size of the ellipse and the absolute age. Hence, we pursue the track of suggesting a cut-off value for Pb*/Pbc in this section, to reduce bias introduced from choice of Pb blank IC and its random variability carried over into the final age interpretation.

5.2 Residual Pb-loss in chemically abraded natural zircon

After concluding in section 5.1 that the Lindskog et al data was troubled by low Pb*/Pbc, they explore here a different tack in that perhaps there were residual domains that had lost Pb that were not removed by the lower temperature etching step employed in this study. But this would have the opposite effect of producing younger ages rather than the older ages they are trying to explain away. L205 states that they conclude that both previous studies "were affected by Pb-loss domains and/or relict inclusions that were not penetrated during chemical abrasion."  This may be true for the Liao et al study that recommended a younger age but not the Lindskog et al data as suggested.
In the available CA studies (e.g. Mattinson et al., 2005; Crowley et al., 2014; Huyskens et al., 2016; McKanna et al. 2024), lower T chemical abrasion (e.g. 180°C or 190°C) results in incomplete Pb-loss removal. Based on this observation we have to make the assumption that the Lindskog et al. (2017) data, which are chemically abraded at 180°C, are affected by this incomplete Pb-loss. It is unreasonable to assume that the higher T chemical abrasion procedure is less efficient than the lower T chemical abrasion. The apparent "old" Lindskog data is potentially an artefact of the blank correction or presence of inheritance. Therefore, we prefer to base our assessment of Pb-loss on the reported duration and temperature of the chemical abrasion, to indirectly assess probable residual Pb-loss. Ultimately, we would state that a concluding assessment of Pb-loss removal in the Lindskog et al. (2017) data is precluded by the lower precision of the data.

5.3 Lead blank isotopic composition correction effects on the spread of zircon U–Pb dates

Here correlations between Pbr and Pbc are considered in terms of their absolute abundances without considering the difference between Pbc as Pb blank or Pb initial and that assuming one or the other would have different implications for the discussion. Again, they fail to discuss the Lindskog et al paper having assigned all non-radiogenic Pb as modern terrestrial Pb and that their interpretation of Pb-bearing inclusions of presumed Ordovician age would have lower 6/4 and 7/4 ratios, producing older calculated ages in both decay schemes. This makes the different in ages between the two studies worse rather than explaining it away. Initial Pb and blank Pb cannot be distinguished by the ID-TIMS method with high confidence (ignoring indirect attempts, as applied to Pbc-bearing phases, as these are to imprecise for high precision zircon U-Pb dating). Arguably, here, the reviewer is too fixated on changing the Pb blank IC in a particular direction. In lines 217-218 we clearly state that it may have an effect that drives either towards younger or older dates.

5.4 The impact of the interpretation strategy on U–Pb zircon ages

This section runs through the different established strategies employed to make a sensible interpretation of a spread of U-Pb zircons commonly found in volcanic ash layers. A combination of issues may be in play here to cause the spread including residence time of zircons (related to magma chamber timescales) and/or detrital grains from an earlier eruptions but issues that have geological explanations. These strategies assume that all the individual ages of zircons in the range are accurate for their respective crystallization and closure. But previous sections have concluded that the 2 previous studies and especially that of Lindskog et al are not reliable ages due to their high Pbc. If they have concluded that at least the Lindskog et al ages are inaccurate then they should not be considered further in this section. Only the zircons with their cut off of >50 for Pbr/Pbc should be considered here. We will outlined more clearly the underlying hypotheses for each of the interpretation strategies to help clarify for the reader. This comparison is necessary to better understand all individual sources contributing to the discrepancy of the Lindskog et al. (2017) and Liao et al. (2020) ages, before conducting a new, revised age calculation. In section 5.5 we do perform the interpretation of data with Pb*/Pbc >50.

5.6 Implications for the Ordovician timescale and the absolute timing of events

This section is eight lines long without discussing the implications for the debate mentioned in the introduction regarding the connection between the Ordovician break up and arrival of the L-chondrite and the GOBE or any implications for the new age on the Ordovician timescale save that it needs to be modified. This should be the most important part of the paper else why bother to get a refined age for this horizon in the first place. As presented, it seems the main justification for this study was to show that that you need higher temperatures and longer times for the chemical abrasion method. As indicated above, we will slightly expand this section and will include additional relevant references, but we maintain that the primary focus of our study has been to explore the significant differences in results/interpretations between previously published studies and to add state-of-the-art U-Pb data to resolve the situation at hand. Thus, we mainly discuss the fundamental implications of our new results, and additional details pertaining to the GOBE and L-chondrite breakup may be found in the cited literature. The 'Likhall' bed is indeed a key level in Ordovician event stratigraphy and time scale, so our results by themselves comprise a significant contribution to move the field ahead.

**Response to reviewer annotations in PDF document:**

Would it make more sense to first describe the rock and its context within the stratigraphy and its relationship to the GOBE then the important feature that it contains zircon - rather than starting with zircons that appear to be named Likhall?

We have rephrased to clarify that the location is named Likhall (line 23).

In different places it is Likhall zircons, Likhall zircon bed, Likhall zircon crystals and Likhall beds - all these terms cannot be correct or at least formally named?

We have simplified the use of Likhall to follow the convention used in Lindskog et al. 2017, now using only Likhall bed and Likhall zircons

This sentence a bit awkward - perhaps trying to say too much in one sentence?

Simplified by removing timescale part (line 26).

In a comparable number of words, it seems possible to make clear the older age suggests no correlation where as the younger age the opposite?

Changing wording as suggested in pdf annotation, favouring -> reaching (line 32).

Not sure that it is mysterious - they must be volcanic in origin. Agreed that it might not be common but i would not think it is mysterious.

Changed wording of mysterious -> uncommon (line 35).

Poor syntax to have "e.g., ..." - make a proper sentence.

Replaced by "for example" (line 37).

But a reference material is quite different from the possible complexity of essentially a detrital zircon population even if primarily volcanic in origin.

Previous work has showcased that volcanic ash zircon can be reproduced at level of 0.1% and sometimes better (take a look at Baresel et al. 2017 for example, where 0.05% was achieved for ash beds marking the Permian Triassic boundary). And in this case, a detrital component appears to be absent.

Chemical abrasion is the method in total - you are referring here to the heating or annealing part of this method.

Clarified to state that the partial dissolution parameter is discussed (line 49-50).

may have

replaced has with may have (line 50).

Line 53 changed plural "procedures" as suggested.

Line 53 changed plural "procedures" and "inclusions" as suggested.

The difference in the two ages should not be referred to as "poor reproducibility" - just different ages outside of their respective uncertainties.

Line 65 changed "poor reproducibility" to "significant difference" (alternative to the suggestion).

In the end, it appears that this may be the explanation for the difference between the Lindskog et al age and the one produced in this paper. Especially since the youngest zircons that were not included in the "plateau age" overlap.

We arrive at a very similar conclusion further on in the manuscript.

Line 73 „using" adopted suggestion.

This section seems out of order - more appropriate in the introductory section to understand the setting?

Sequence adjusted to position 3.

Poor syntax to have "e.g." mid sentence. Not required here.

Adopted, removed „e.g.".

Line 145 "been" inserted as suggested.

Not clear how the color coding shows that the majority of analyses plot on or near concordia - the location of the ellipses shows this, not the color coding which shows Pb*/Pbc ratios.

Color coding has been revised to make it clearer. Reduced color scale to 3 and grouped based on Pb*/Pbc <15, 15-50, >50 consistent with revised text discussion.

Line 150 "the" deleted as suggested.

As stated, confusing. Perhaps better to state something like "A previous study of zircon systematics has shown that accuracy of 206/238 ages improves significantly with Pbr/Pbc ratios greater than 15 (ref)."

Rephrased to make clear that Pb*/Pbc greater than 15 is critical. Line 153.

why is "at very variable 206/238 dates" necessary here?

Removed very. Line 156.

Should be stated what the cutoff number is here.

Added cutoff value >50. Line 157.

"ideally" yes, but perhaps more honest to state that "If correct, higher chemical abrasion temperatures may have dissolved these Pbc bearing phases".

Line 160. Rephrased to make it conditional on the hypothesis that inclusions control the Pbc.

Line 206 "low blank or" deleted as suggested.

Line 206 "define" insertion rejected, not suitable in our opinion.

Line 206 Why 50 now? Above it was 15-20. Clarified through edits in the paragraph above.

Line 207 Above you state 6 pg - rather than 5.99 pg. The rounded value makes the point well enough? Rounded to 6 now as above.

Line 209 It is not mentioned what the effect of the low Pbr/Pbc would have in terms of the distribution of zircons on a concordia plot. That is the relative effect on the 206/238 age relative to the 207/235 age if an inappropriate Pbc IC was used. And the effect on the

uncertainties as the common Pb correction is propogated. Added brief explanation that correction may result in younger or older dates and refer to section 5.3 where this is discussed.

Line 223 This seems a bit late to be explaining the chemical abrasion method when you have already assumed the reader knows about this in the sections above. Removed introductory sentence and modified second sentence to start the paragraph.

Line 226 What is an "analytically concordante result"? Is this different from a "concordant result"" removed analytical (redundant).

Line 227 Again, more basic background information that is assumed above to be known by the reader. Removed redundant sentence.

Line 235-237 But how does anyone know that this is related to true variations in zircon ages due to residence times rather than slight Pb loss? Make clear if this is known. We clarify that variations are not fully understood in a second sentence now (Schaltegger et al., 2021).

Line 239-242 Is this true for the 1 Myr difference between the Lindskog et al age and the age presented here? Or would they be "moving" along a discordia line that would be indistinquishable from concordia at this scale?

Sentence modified to clarify that ages may move along concordia (see Condon et al. 2024) and that the older Lindskog data do not permit this evaluation.

Line 248 Rather than being curious, is it not the wrong interpretation of why this data is on average older than both your new data and that of Liao et al. In the same sentence we already point the reader to the next section, where blank correction effects are discussed.

Line 251-252 See point above as to whether you would see discordance with the errors as stated. The following sentence addresses this in the original submission.

Line 251 What do you mean by "technically discordant"? removed technically (redundant).

Line 252 What do you mean by "normal discordance"? Below concordia in the sense of the opposite of reverse discordance? Reworded to Negative discordance, although this is a term commonly understood in the community.

Line 254 But how does this explain the older ages of Lindskog et al? It seems Pb-loss domains that were not removed can only cause younger ages? Clarified that this may be deduced from higher Pbc and/or negative discordance.

Line 259 If you are going to explain the older ages of Lindskog et al as being Pbc related, why present it above as being related to discordance due to residual Pb-loss domains. Very confusing where this is going. Discussion of discordance should be included for completeness, any ID-TIMS U-Pb practitioner will be aware that Pb-loss and blank correction both affect a calculated date. It can however not be assumed that the lower T dissolution is more effective than the higher temperature in removing Pb-loss. Thus, if residual Pb-loss is present at higher temperature chemical abrasion, it must be in the Lindskog data too.

Line 261 This is not technically correct - it may be desireable but not strictly necessary if the Pbr/Pbc is high enough. The correction for non-radiogenic Pb will have an uncertainty that is part of the final age uncertainty. The placement of the comment by the reviewer is not clear

here. The sentence it is attached to clearly states the assumption that no initial Pb is incorporated into zircon during formation (Watson 1997). We

Line 261 Recommend sticking with just initial Pb if that is what you are talking about. Common Pb in many peoples mind is anything that is not radiogenic - either initial or blank. We removed "common Pb" to simplify.

Line 263 This is also misleading. If the Pb blank is consistent, it can be subtracted and then the residual 204Pb is assigned to initial Pb with an assumed IC. This Pbi may not be in the zircon but rather in inclusions. But it is not correct to say that all 204Pb is necessarily assigned to blank Pb. The uncertainty on the blank amount, blank IC and the IC for the Pbi are all part of the final errors when calculated by Schmitz and Schoene, 2007. In zircon U-Pb ID-TIMS dating, it is standard practice to correct all initial Pb as blank Pb, which we strictly follow here. We may of course perform a blank and common Pb correction for common Pb bearing phases such as titantite, rutile and apatite, but this would be unconventional for zircon. What the reviewer is suggesting would introduce additional speculation and assumptions, which we object to do. We also have no estimation of the IC of the Pb in the potential inclusions.

Line 266 Also incorrect to state that accuracy in defining the IC of blank Pb increases the precision of the U-Pb ages. The precision of the final ages depends on the uncertainties assigned to the blank Pb IC, not their accuracy. Clarified in the text that "accurate and precise" blank correction is required to arrive at accurate and precise dates.

Line 267 Also incorrect. A low Pbr/Pbc ratio does not inherently make the ages too old or too young if the Pbc is properly assigned - in this case the age will be accruate but the precision poor. In this sentence we discuss "inaccurate" correction of blank and its effect on calculated dates. The sentence above discusses the scenario of "properly assigned" blank correction. We make no changes.

Line 269 This is correct that Lindskog et al assigned all 204Pb to blank - but this does not mean this is necessary, as implied in the previous paragraph. Attributing all 204Pb to blank is common practice in ID-TIMS U-Pb dating of zircon.

Line 273 scattered. Adopted in the text.

Line 287 There appears to be confusion about the origin of the non-radiogenic Pb (or Pbc). As correctly pointed out, Lindskog et al assigned all Pbc as  blank Pb and subtracted 206 and 207 relative to 204 using assumed blank IC. But here they are suggesting that the Pbc is Pbi in inclusions which would not have blank IC but that of Ordovician Pb with a lower 6/4 and 7/4 ratio and, therefore, less 206 and 207 subtraction and older ages. I cannot see this complexity discussed anywhere. In the absence of knowledge or reasonable estimation of the potential inclusions Pb IC we cannot reasonably consider subtracting blank and inclusion IC separately. The standard practice in ID-TIMS U-Pb dating is to attribute all initial Pb to blank.

Line 296 dates added plural.

Line 305 This seems to be mixing a discussion of strategies of interpreting a range of accurate zircon ages with having a range of zircon ages that are inaccurate. These strategies listed are typically intended to address a range of zircon ages from a volcanic bed where the individual ages are assumed to be accurate. They will not help with inaccurate ages due to the improper assignment of Pbc. The results of our exercise show that comparability improves when

exercising same interpretation strategies to the three data sets. We think the reviewer is wrong in his assessment here.

Line 305 Where would "old radiogenic Pb" be coming from? Added that this might derive from inherited cores.

Line 306 If you have concluded above that the ages of Lindskog et al are inaccurate due to an inappropriate assignment of Pbc, then any further discussion of these zircon ages and how they relate to the new age seem irrelevent. The model results disprove the reviewers statement in our opinion.

Line 327 Add units of Myr for these delta t estimates. Done, added Ma.

Line 344 The blank Pb IC is given in Lindskog et al in footnotes to the data table. Removed sentence.

Line 367 deleted some as suggested.

Line 368 so as changed to "given that" as suggested.

Line 377 Why is this age not explained? Age is discussed in the references mentioned in the text.

Line 385 This list of a-c does not follow on from the opening line. Sequence has been adjusted based on all reviewer comments.

Line 449 Missing Liao et al reference. Added missing reference.

Response to Reviewer #3

Thanks to the editor and authors for the opportunity to review this interesting paper. This study revisits a previously dated zircon-bearing carbonate bed, the Likhall bed, which has important implications for the Ordovician time scale but has yielded inconsistent dates between studies. The authors attribute these differences to both

• methodological issues (mainly incomplete chemical abrasion by previous studies, as well systematic differences in tracer calibration between studies using an ET spike and a study using an in-house spike) and

• interpretive differences relating to how one determines a single age for a bed that contains multiple dated zircon grains.

The authors grapple with these problems and put forward a new weighted mean age estimate

for the bed. The study is well-written, concise, and clear. I support publication of the study and make the following (hopefully constructive) suggestions.

(AC) Dear Reviewer, thank you for taking the time to read and comment on the submission. We appreciate your opinion and constructive comments. Below we respond to your comments.

This bed was previously dated by two studies, Lindskog et al. 2017 and Liao et al. 2020, and ages from all three studies are shown in Figure 2.

- This figure shows all the ages with their X uncertainty, but I believe they should all be shown with their Y uncertainty (including tracer calibration uncertainty), as the Liao data were determined using an in-house spike and the Lindskog/Paul data were generated using an ET spike. If the authors don't wish to or can't do this, then I suggest caveating this in the caption.

The expansion to the Y uncertainty will inflate the bars by ca. 3 kyrs (ET datasets), which is too little to be visible/significant. Similarly for the Liao et al. data.

Line 95: Can the authors provide a citation supporting the assumption of a magma initial Th/U ratio of 3.5 +/- 1?

This is an assumed value, within a range of values commonly used.

Figure 1:

- I suggest recoloring this figure so that the cut-off value of 50 is at an inflection point in the colorscale and/or indicate on the colorscale where 50 is. It is hard to look at this and immediately understand that some of the green grains that overlap with the concordia curve are actually below 50 and should be left out.

- It's also somewhat confusing that the grains "selected" for analysis here are not the same as the grains selected in Table S1, column "interpretation strategy i."

  o I suggest annotating this column in Table S1 to indicate the three grains that are selected for analysis using the more restrictive Pb*/Pbc screen.

  o I suggest adjusting the figure so that all grains that are "selected" using

  "interpretation strategy i" have a black dashed ellipse and those that are "selected" using the more strict Pb*/Pbc screen have a black ellipse (as they do now).

- Isn't a part of the selection criteria also that the ages are within the youngest cluster (line 305)? This should be stated in the caption.

We have made the following adjustments to figure 1: Colour gradient modified to 3 bands of Pb*/Pbc (>15, 10-50 and >50), which are also utilized in the text and the final age interpretation. Table S1 amended to include the Pbc/Pb* band color coding (grey, blue and orange). As for adding strategy i), this overloads the figure in our opinion. Expanded the figure caption to clarify the interpretation strategy.

Line 125: I think the geological setting should be given greater priority within the manuscript and encourage the authors to include a figure showing the field context of the bed sampled. I

understand that this has been described elsewhere, but such a figure would increase the value of this manuscript to the reader. Similarly, images of the zircon grains have been previously published elsewhere, but including some images in this manuscript or in the supplement would be helpful.

The placement of the geological setting is adjusted. We have found no meaningful correlation with zircon petrography and U-Pb systematics.

The authors first consider differences in the chemical abrasion techniques used between studies, and argue that incomplete chemical abrasion of inclusions and metamict zones within zircons resulted in inaccurate ages for the previous studies. Previous studies used lower temperatures for their chemical abrasion procedures than the current study. The main data the authors use to support this idea is shown in Figure 4, which plots Pbc vs Pb* for each of the three studies.

• This is more of an aesthetic suggestion, but I suggest making all the plots the same size

and aligning their left y-axis.

Figure axes have been aligned.

• I also think that plotting all the data with the same axes or on the same plot would help the

authors make their point that their approach has resulted in much smaller Pbc

measurements and presumably more precise and accurate dates.
We have initially opted against this since the scale difference is almost a factor of 10 when all data are displayed. We would like to maintain the current style but thank for the reviewer for their consideration and constructive comment.

• Because chemical abrasion is meant to remove zones of zircon affected by Pb loss, it's

easy to follow this argument for the Liao data, which the authors argue is too young because

of incomplete chemical abrasion, but harder to follow it for the Lindskog data, which is too

old (and the authors discuss this). They suggest that the "too old" ages come from

inaccurate and imprecise blank corrections.
o Line 231: "If we assume that larger zircons contain more Pb* and a larger volume

of Pbc-bearing inclusions…" I'm happy to agree that larger zircons would have

more Pb* and Pbc, but I'm missing a step in the logic of the sentence here. I think

it's that larger zircons with more Pbc would be more impacted by an erroneous

blank correction, but perhaps it can be spelled out a bit more clearly.

We have clarified the sentence a bit more.

The authors consider several different interpretation strategies. They identify five ways that one might determine the age of this bed:

1. Weighted mean of a subset of data

2. Youngest cluster of overlapping ages at 2-sigma

3. Use entire range of concordant zircon analyses as autocrystic growth in magma chamber

4. Use the youngest concordant grain as the best proxy for the timing of eruption

5. A Bayesian approach as suggested by Keller et al. 2018

• I would rephrase 5 to something like "A Bayesian approach such as those suggested by Keller et al. 2018 or Traylor et al. 2021."

We considered the comment by the reviewer but do not want to expand the Bayesian model section in this short article, as we believe significant discussion would be needed to truly be meaningful, going beyond our scope.

o The authors immediately state that #5 is outside the scope of this study, but I don't agree. The approach used by Keller et al. 2018 is relatively easy to implement and a publicly-available Jupiter Notebook provides help at https://github.com/brenhinkeller/Chron.jl?tab=readme-ov-file. I suggest that it be included and discussed; if the authors have a major disagreement with this approach, they should show why.

The Keller model is available indeed and calculations may be performed quite easily. However, comparing the three data sets is strictly not feasible. To achieve comparable results, all data sets must have eliminated Pb-loss efficiently, be directly comparable in instrumental setup and of sufficient high precision. This is not possible here. The Keller model will generate an model age that approximates the youngest zircon U-Pb age of the data set. Thus, when rejecting only analyses that a clearly discordant in Concordia space, the Keller model results will be:

|  | Age | 2sigma |
|---|---|---|
| Liao | 464.9969 | 0.360647 |
| Lindskog | 466.7484 | 0.803836 |
| this study | 466.4687 | 0.261105 |

We thus need to develop a strategy to unify the data before we can expect comparable results. This, in turn, requires a much larger dataset first, which goes beyond the scope of what we

want to discuss here.

o Traylor et al. refers to modifiedBChron, which would consider all analyses as part of a summed probability density distribution function and then use stratigraphic superposition in a Bayesian model to determine an age that is supposed to be a better representative of geological uncertainty. Using only the new data produced by this study, of course this can't be applied as we don't have the benefit of stratigraphic superposition, but if, as the authors state, there are other dated bentonites in the section with clear stratigraphic relationships, why not give it a try, and see how it compares? modifiedBChron is also fairly easy to implement. I suspect the outcome will mostly highlight the importance of getting good age constraints elsewhere in the section, but this would be worth highlighting too.

The Bayesian approaches introduced by Traylor and Keller are exciting new developments that help us understand what the meaning of a dated ash bed means in stratigraphic context.

We thank the reviewer for the very constructive thought and consideration. We will be happy to explore the Traylor et al. model in a second research item, to not extend beyond the short communication format chosen here. We will like to explore this when presenting some new bentonite dates from Argentina, that temporally overlap with this work, where we have supporting fossil records and stratigraphy.

This paper's ambition is to make the point that the Ordovician timescale must be reconsidered, and it certainly provides evidence in that direction, but at present, this paper misses an opportunity to grapple with these emerging approaches and their implications for the timescale.

We thank the reviewer and hope to present a different, second, manuscript to the community that may incorporate more of the above mentioned points. Here, we have opted to focus primarily on more technical and interpretational aspects to clarify the U-Pb zircon date of this layer.

The authors spend a lot of time discussing using Pb*/Pbc as a screening metric and note that it significantly improves accuracy while coming at a significant cost of greatly reducing the number of grains that are viable. Can they give non-zircon geochronologists some sense of how a strict requirement of Pb*/Pbc > 50, for example, would affect the universe of published CA-ID-TIMS zircon data? Would it knock out >50% of published zircon grains, as it does here? Given the expense and time-consuming nature of CA-ID-TIMS analysis, how do they recommend other workers grapple with this?

Continued progress in laboratory procedures and technical improvements has been a fundamental pillar of the CA-ID-TIMS U-Pb community for the past decades now, and we expect that future data sets will continue that trend. As we gather more and more data on how to avoid common Pb (inclusions and/or consumables), newly generated data are expected to fall above that threshold. For past data, certainly a large number will become less relevant.

How this data may be considered in the future would need to be evaluated by the community in our opinion.

Line 299: "leaves only 8 of 22"—yes and worth stating explicitly that these 8 are not all within the youngest age cluster.
Emphasized more and clarified that only 3 grains form the weighted mean age.

The authors describe some of the changes to the Ordovician timescale that their new age requires.

• It would be helpful for the authors to visualize these changes in a "before" and "after"

figure.
That is something that we hope to describe more extensively in a follow up submission, when more data is generated and combined from relevant sections.

• The authors also note that the timing of L-chondrite breakup should be revised to c. 467.1

Ma, but the reader isn't given enough context about why this date should be assigned to

the breakup, (which isn't the age they give the Likhall bed).
We have expanded this section and give more references to works which have investigated the issue.

• As the authors motivated this study by mentioning the controversial hypothesis linking

meteorite bombardment with Ordovician biodiversification in the introduction, they should

return to this hypothesis and discuss the alignment or un-alignment of these two events in

light of their new results.

We have extended the section slightly and added more references for the reader.

Finally, the references:

• Liao et al. 2020 is a key study for this paper and the source of much of the legacy data

discussed, but it does not appear in the reference list. This must be addressed before final

publication.

Added the missing reference.

• I suggest reformatting the Schmitz et al. citation in the references to superscript 205, 235,

238, etc. Same thing for the von Quadt et al. citation and 10^13.

Modified the references accordingly.

• On the topic of CA-ID-TIMS dating of cryptotephra from carbonates: The authors might

be interested to know about a similar study focused on retrieving volcanic zircon from

carbonate rock: Finzel and Rosenblume 2021, Geology. I think referencing this study

would help make the authors' point that determining best interpretive practices for

carbonate-derived crypotephra ashes is a question with applications beyond the single bed

considered here.

We thank the reviewer for pointing out this reference, we will check it and incorporate if suitable.